# Thought Communication in Multiagent Collaboration

**Yujia Zheng**[1,2]    **Zhuokai Zhao**[2]    **Zijian Li**[3]    **Yaqi Xie**[1]
**Mingze Gao**[2]    **Lizhu Zhang**[†,2]    **Kun Zhang**[†,1,3]

[1] CMU    [2] Meta AI    [3] MBZUAI
{yujiazh, kunz1}@cmu.edu   {zhuokai, lizhu}@meta.com

## Abstract

Natural language has long enabled human cooperation, but its lossy, ambiguous, and indirect nature limits the potential of collective intelligence. While machines are not subject to these constraints, most LLM-based multi-agent systems still rely solely on natural language, exchanging tokens or their embeddings. To go beyond language, we introduce a new paradigm, *thought communication*, which enables agents to interact directly mind-to-mind, akin to telepathy. To uncover these latent thoughts in a principled way, we formalize the process as a general latent variable model, where agent states are generated by an unknown function of underlying thoughts. We prove that, in a nonparametric setting without auxiliary information, both shared and private latent thoughts between any pair of agents can be identified. Moreover, the global structure of thought sharing, including which agents share which thoughts and how these relationships are structured, can also be recovered with theoretical guarantees. Guided by the established theory, we develop a framework that extracts latent thoughts from all agents prior to communication and assigns each agent the relevant thoughts, along with their sharing patterns. This paradigm naturally extends beyond LLMs to all modalities, as most observational data arise from hidden generative processes. Experiments on both synthetic and real-world benchmarks validate the theory and demonstrate the collaborative advantages of thought communication. We hope this work illuminates the potential of leveraging the hidden world, as many challenges remain unsolvable through surface-level observation alone, regardless of compute or data scale.

## 1 Introduction

Natural language has enabled human collaboration at scale, but it also imposes fundamental limitations. While powerful, language is inherently sequential, ambiguous, and imprecise, offering only an indirect and fragmented reflection of thought [von Humboldt, 1988]. This constraint is deeply rooted in human cognition, which lacks direct channels for transmitting mental content. Machines, however, are not subject to the same physical constraints of speech or perception. This difference may be one of the central reasons why superhuman intelligence is possible. Every transformative achievement, from scientific discovery to societal progress, relies on collaboration. Likewise, superhuman intelligence will require not only individual reasoning beyond human capability but also collective reasoning beyond human coordination [Vinge, 1993]. This calls for a new form of communication that transcends the limits of language.

However, existing large language model (LLM)-based multi-agent systems (MAS) rely on natural language as the medium of communication, exchanging information via tokens or their embeddings [Du et al., 2023, Liang et al., 2023, Pham et al., 2023, Zhang et al., 2024a, Zeng et al., 2025, Wang et al., 2025b]. These systems typically assume that multiple LLM agents exchange natural language messages to convey internal ideas and coordinate toward a shared goal. However, natural

---

[†]Equal advising.

language remains fundamentally limited in its ability to express the underlying latent thoughts that drive reasoning and decision making. As a result, current systems remain restricted by the bottlenecks of language, limiting their potential for superhuman collaboration. Indeed, recent empirical analyses [Cemri et al., 2025, Hu et al., 2025] highlight that many failures in inter-agent collaboration stem from vague message specification and inter-agent misalignment, both ultimately caused by the indirect nature of lossy language-based communication. Then, the core question reveals itself:

*What form of communication goes beyond the limits of language?*

To answer this, we turn to the idea of communication through latent *thoughts*. Nothing is more direct than transmitting what one truly thinks, i.e., *telepathy*. Just as human actions are guided by internal mental states, agents likely operate based on latent representations that encode goals, beliefs, and reasoning. If these could be identified, agents could share them directly, bypassing the ambiguity and distortion of language. This enables a fundamentally different mode of communication, based not on the exchange of surface tokens or their embeddings, but on the direct transfer of intent and understanding. Furthermore, in multi-agent settings, some thoughts are intended to be broadly shared, while others are inherently private or uniquely tailored to certain individual agents. Revealing both the latent thoughts and their structural organization allows agents to better detect alignment, resolve conflicts, and integrate diverse reasoning paths.

**Contributions:** We formalize this idea by introducing a latent generative model for inter-agent communication. Specifically, we assume that the model states $H_t$ of all agents before communication round $t$ are generated from a set of latent thoughts $Z_t$ through an unknown function $f$, such that $H_t = f(Z_t)$. We establish both a nonparametric identifiability result that guarantees recovery of latent thoughts, and a general framework that facilitates direct mind-to-mind communication.

*Theoretically*, we prove that in a general nonparametric setting, both shared and private latent thoughts can be identified from hidden states under a sparsity regularization. Our identifiability result ensures that the recovered latent representations reflect the true internal structure of agent reasoning. Moreover, we show that the structures between thoughts and individual agents can be reliably recovered, enabling a provable correspondence between agents and their cognitive content. Experiments on various synthetic environments confirm the validity of the theory.

*Practically*, we develop a principled framework for latent communication among agents. Guided by the theory, we implement a sparsity-regularized autoencoder to extract latent thoughts from agent hidden states and infer the underlying mapping between agents and these thoughts. Each agent is equipped with a set of inferred thoughts, along with the structure of how each thought is shared. This allows agents not only to understand what others are thinking but also to reason about which thoughts are mutually held or privately maintained. Experiments across diverse models and scenarios demonstrate that communication beyond language directly benefits collaboration among LLM agents.

## 2 Problem Formulation

In this section, we formalize the data-generating process behind agent responses, providing the foundation for our theoretical analysis.

**Data-generating process.** We illustrate the data-generating process in Fig. 1 and formalize it as:

$$Z_t \sim P_z, \quad H_t = f(Z_t), \quad (1)$$

where $Z_t = (Z_{t,1}, \ldots, Z_{t,n_z}) \in \mathbb{R}^{n_z}$ denotes the latent *thoughts* of agents at communication round $t$, and $Z_{t,i} \in \mathbb{R}$ for $i \in [n_z]$ represents a latent variable denoting a single thought. Let $n_a$ be the number of agents, at communication round $t$, the global model states[*] of all agents are given by

$$H_t = (H_t^{(1)}, \ldots, H_t^{(n_a)}) = (H_{t,1}, \ldots, H_{t,n_h}), \quad (2)$$

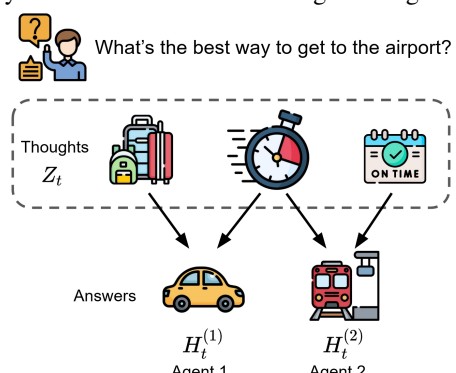

Figure 1: Each agent answers the same question by selecting a subset of latent thoughts $Z_t$. Agent 1 chooses a *car* 🚗 based on *carrying luggage* 🧳, while Agent 2 selects a *train* 🚆 for *schedule punctuality* 📅. Both share the thought of *speed* ⏱️.

*What's the best way to get to the airport?*

Thoughts $Z_t$

Answers

$H_t^{(1)}$ Agent 1    $H_t^{(2)}$ Agent 2

---

[*]We refer to this as the *model* state instead of *hidden* state to avoid confusion with the *latent* thoughts $Z_t$. Specifically, model state corresponds to the hidden layer representation of the underlying fundation model.

where each $H_t^{(j)} \in \mathbb{R}^{n_{h_j}}$ summarizes the model states of agent $A_j$ prior to the communication round $t$, and $n_h = \sum_{j \in [n_a]} n_{h_j}$. The mapping from latent thoughts to hidden states is governed by an unknown generating function $f$, assumed to be invertible (to preserve information) and twice differentiable (to ensure well-defined gradients), following the literature [Hyvärinen et al., 2024].

***Example*** 1. Fig. 1 illustrates the data-generating process. In response to the question *What's the best way to get to the airport?* a set of latent thoughts $Z_t$ is considered, including factors such as carrying luggage, speed, and punctuality. These thoughts, represented as latent variables $Z_{t,i}$, are mapped through the generating function $f$ to produce each agent's answers, which are summarized by their model states $H_t^{(j)}$. For example, Agent 1 emphasizes thoughts related to *luggage* 🧳 and *speed* ⏱️, resulting in the state $H_t^{(1)}$ that leads to choosing a *car* 🚗. Agent 2, influenced by *speed* ⏱️ and *schedule punctuality* 🗓️, forms the state $H_t^{(2)}$ and selects a *train* 🚆. This example illustrates how the underlying process $f$ encodes shared and private latent thoughts into agent-specific responses.

**The structure of thoughts.** While prior strategies have focused on communication through language or token embeddings, we propose a fundamentally different paradigm where agents share latent thoughts directly. To achieve this, we propose a communication paradigm in which agents access relevant latent thoughts instead of surface-level messages or embeddings. Rather than exposing all latent thoughts $Z_t$ to every agent uniformly, we focus on learning the structure of the revealed thoughts so that each agent receives only the most relevant information to its goals and role. This requires modeling how thoughts are selectively shared, as some may represent common knowledge useful to many agents, others may be specific or private to individual goals, and some may be irrelevant or even distracting to certain agents.

We formalize the structural dependency between latent thoughts $Z_t$ and model states $H_t$ through the non-zero pattern of the Jacobian $J_f(Z_t)$, represented as a binary matrix indicating which components of $Z_t$ influence which components of $H_t$:

$$B(J_f) \in \{0,1\}^{n_h \times n_z}, \quad B(J_f)_{i,j} = \begin{cases} 1 & \exists z_t \in \mathcal{Z}_t, J_f(z_t)_{i,j} \neq 0, \\ 0 & \text{otherwise.} \end{cases} \tag{3}$$

The model state of each agent $A_k$ is represented as a slice $H_t^{(k)} = (H_{t,k_l}, \ldots, H_{t,k_h})$, where $k \in [n_a]$; and $\{k_l, \ldots, k_h\}$ denotes the index range in $H_t$ corresponding to agent $k$. We define the set of latent thoughts relevant to agent $A_k$ as

$$Z_{H_t^{(k)}} := \left\{ Z_{t,j} \in Z_t \mid \exists i \in [k_l, k_h] \text{ such that } B(J_f)_{i,j} \neq 0 \right\}. \tag{4}$$

In other words, $Z_{H_t^{(k)}}$ consists of all latent thoughts that influence at least one component of agent $A_k$'s hidden state, as determined by the non-zero pattern of the Jacobian $B(J_f(Z_t))$.

## 3 Identifiability Theory

Before leveraging thought for communication, a critical question arises: how can we ensure that the recovered thoughts correspond to the true ones underlying agent responses? To address this, we establish an identifiability theory for reliably recovering the latent thinking process. We begin with the *identification of the latent thoughts* (§3.1 and §3.2), then explore *the structure between thoughts and agents* (§3.3). All proofs are included in Appx. A.

### 3.1 Identifiability of *Shared* Thoughts

Communication often begins with establishing common ground, which typically requires confirming shared beliefs before addressing disagreements. If the shared part of the latent thought can be reliably disentangled from other components, then communication can start from a faithful common basis. Our identifiability result guarantees this: by recovering shared latent variables that are not entangled with any others, we ensure that inter-agent communication is grounded in true cognitive overlap.

We first introduce some additional technical notations. We define the support subspace $\mathcal{S}_{J_f}$ as the set of matrices $S \in \mathbb{R}^{n_h \times n_z}$ whose nonzero entries are restricted to the nonzero pattern of $J_f(Z_t)$:

$$\mathcal{S}_{J_f} := \left\{ S \in \mathbb{R}^{n_h \times n_z} \mid B(J_f)_{i,j} = 0 \Rightarrow S_{i,j} = 0 \right\}. \tag{5}$$

We further denote $M$ as a matrix with the same nonzero pattern of $m(Z_t)$ in $J_f(Z_t)m(Z_t) = J_{\hat{f}}(\hat{Z}_t)$, and write $\overset{d}{=}$ to denote equality in distribution.

**Theorem 1** (Identifying the shared thoughts). *Suppose that for each $i \in [n_x]$, there exist points where the Jacobians $J_f(Z_t)_{i,\cdot}$ span the support subspace $\mathcal{S}_{J_{f_{i,\cdot}}}$, and that $(J_f(Z_t)\mathcal{M})_{i,\cdot} \in \mathcal{S}_{J_{\hat{f}_{i,\cdot}}}$ at those points. If $H_t \stackrel{d}{=} \hat{f}(\hat{Z}_t)$ for a model $(\hat{f}, \hat{Z}_t)$ following §2 with $\ell_0$ regularization on $J_{\hat{f}}$, then for any pair of agents $A_i$ and $A_j$ at round $t$, there exists a permutation $\pi$ over $[n_z]$ such that $\frac{\partial Z_i}{\partial \hat{Z}_{\pi(j)}} = 0$ for any $Z_i \in Z_{H_t^{(i)}} \cap Z_{H_t^{(j)}}$ and any $Z_j \in (Z_{H_t^{(i)}} \cup Z_{H_t^{(j)}}) \setminus (Z_{H_t^{(i)}} \cap Z_{H_t^{(j)}})$.*

**Interpretation and discussion.** Intuitively, Thm. 1 ensures that, up to permutation, the recovered shared thoughts between any pair of agents are disentangled from all other latent variables in the system. The permutation reflects the standard relabeling indeterminacy common to identifiability results [Hyvärinen et al., 2024, Moran and Aragam, 2025]. For instance, in Fig. 1, we can make sure that the recovered thought *speed* 🕙 will not be mixed with others such as *luggage* 🧳 or *schedule punctuality* 📸. Without this guarantee, any recovered thought can be a mixture of any other thoughts, since the unknown generating function $f$ is essentially a mixing procedure. Thus, this disentanglement implies the recovery of the target shared components, provided that the generating function is invertible and thus information-preserving. This has practical implications: given any group of agents, we can decompose them into pairs, each yielding identifiable shared thoughts. By composing the recovered components across different pairs, we reconstruct the common cognitive basis and reveal how thoughts are distributed across agents, including the degree of agreement, which is essential for enabling trustworthy and informative latent communication.

**Assumption.** The assumption has been widely adopted in the identifiability literature [Lachapelle et al., 2022, Zheng et al., 2022], which eliminates degenerate cases where the population is too limited for the Jacobian to even reflect the dependency structure. It requires that the generating function $f$ varies sufficiently across the population so that there exist several points for the Jacobian to span the support subspace $\mathcal{S}_{J_{f_{i,\cdot}}}$. Requiring $(J_f(Z_t)M)_{i,:} \in \mathcal{S}_{J_{\hat{f}_{i,\cdot}}}$ holds at these points is also mild due to $(J_f(Z_t)m(Z_t))_{i,\cdot} = J_{\hat{f}}(\hat{Z}_t)_{i,\cdot}$, especially in the asymptotic regime where identifiability is defined.

### 3.2 Identifiability of *Private* Thoughts

In Thm. 1, we established the identifiability of shared thoughts, providing a guarantee for recovering the underlying common ground between agents. However, effective collaboration is not solely about enforcing consensus or resolving disagreements. In fact, homogeneity can be counterproductive in the long term [Prat, 2002]. Just as humans value cognitive diversity as a source of novelty and innovation, different agents may contribute unique perspectives that are essential for solving complex tasks. For instance, in a collaborative planning task, one agent may recognize rare constraints based on its prior experience that others overlook. Preserving such private thoughts can lead to better overall solutions through complementary reasoning. Motivated by this, we now extend our theoretical analysis to show that private thoughts can also be identified:

**Theorem 2** (Identifying the private thoughts). *Suppose the assumption in Thm. 1 holds. If $H_t \stackrel{d}{=} \hat{f}(\hat{Z}_t)$ for a model $(\hat{f}, \hat{Z}_t)$ following §2 with $\ell_0$ regularization on $J_{\hat{f}}$, then for any pair of agents $A_i$ and $A_j$ at round $t$, there exists a permutation $\pi$ over $[n_z]$ such that $\frac{\partial Z_i}{\partial \hat{Z}_{\pi(j)}} = 0$ for any $Z_i \in Z_{H_t^{(i)}} \setminus Z_{H_t^{(j)}}$ and any $Z_j \in Z_{H_t^{(j)}}$.*

**Interpretation and discussion.** Similar to Thm. 1, Thm. 2 adopts a pairwise perspective and provides guarantees for recovering the hidden private thoughts of any given agent. Specifically, for any pair of agents, it shows that the private component of either agent can be disentangled from all remaining latent variables. For instance, in Fig. 1, recovered latent variables corresponding to the thought *being able to carry luggage* 🧳 – which may explain Agent 1's choice of *car* 🚗 – is not entangled with unrelated thoughts like *speed* 🕙 or *schedule punctuality* 📸, which influence Agent 2's preference for the *train* 🚆. Without such disentanglement, we risk misattributing the decision to an incorrect or irrelevant latent cause, leading to misalignment in communication.

This again implies that, under invertibility, the true private thoughts can be recovered. By composing the results across different agent pairs, we can infer how agent-specific a given thought is. For example, by analyzing all pairwise decompositions in a large group, we can identify thoughts that are truly unique to individual agents, capturing insights that would otherwise be lost due to their rarity or lack of popularity. This connects naturally to the classical long-tail phenomenon: some thoughts may

be infrequent, but they carry critical value. Our theory ensures that these less common but meaningful components are not discarded, enabling inclusive communication and collaboration among agents.

## 3.3 The Structure of Thoughts

Having established the identifiability of both shared and private thoughts, we now turn to a deeper question: how are these thoughts structurally organized across agents? That is, beyond identifying each thought, can we also identify which agents hold which thoughts? In many scenarios, especially those involving coordination, it is not enough to only know the content of internal reasoning. We must also know how that reasoning is distributed across individuals. We formalize this in Thm. 3:

**Theorem 3** (Identifying the structure of thoughts). *Suppose the assumption in Thm. 1 holds. If $H_t \overset{d}{=} \hat{f}(\hat{Z}_t)$ for a model $(\hat{f}, \hat{Z}_t)$ following §2 with $\ell_0$ regularization on $J_{\hat{f}}$, then the nonzero pattern $B(J_{\hat{f}})$ is identifiable up to relabelling, i.e., $B(J_{\hat{f}}) = B(J_f)P$ for a permutation matrix $P$.*

**Interpretation and discussion.** Thm. 3 establishes that the structure linking latent thoughts to agents' internal states is identifiable up to permutation. In other words, we can recover not only the content of each thought, but also determine which agents hold which thoughts, and which thoughts are shared. Returning to Fig. 1, this means we can infer that both agents care about *speed* 🕙 (shared), while only Agent 1 emphasizes *carrying luggage* 🧳 (private) and only Agent 2 prioritizes *being on time* 📆 (private). This structure-level recovery is crucial: it enables agents to assess not just what others are thinking, but also how similar or different their internal reasoning is, supporting more informed and adaptive communication. In practical terms, this guarantees that agents can identify points of alignment and divergence without confusion. When scaled to larger systems, this enables the reconstruction of a full thought-agent incidence structure, revealing clusters of agreement, regions of conflict, and sources of novel inputs. Such structural insights are foundational for building systems that coordinate robustly and interpret each other's intentions with precision.

## 3.4 Discussion on Theoretical Contribution

To the best of our knowledge, this work is the first to consider the latent generative process underlying LLM agent responses and to provide identifiability guarantees for recovering latent thoughts. Beyond its novelty in the multi-agent LLM setting, Thms. 1, 2, and 3 also present a new contribution to classical identifiability theory. Prior work typically focuses on recovering all latent variables (up to standard indeterminacies), with assumptions that go beyond the basic setup that we adopt, such as access to weak supervision [Hyvärinen et al., 2019, Khemakhem et al., 2020], specific function classes [Taleb and Jutten, 1999, Buchholz et al., 2022], or structural criteria on the dependency graph [Moran et al., 2021, Zheng et al., 2022].

In contrast, our approach takes a completely different route. Instead of aiming for global recovery, we focus on pairs of observed variables (agents) and seek to recover as much hidden information as possible from them. Since we rely only on basic assumptions and do not use the additional constraints or auxiliary signals commonly adopted in the identifiability literature, full recovery of all latent variables is known to be impossible. Therefore, we target a coarser perspective that is still meaningful for communication, such as the shared/private thoughts disentangled by our theorems. This is not only practically useful but also theoretically important, as previous methods with global conditions offer no guarantees when their assumptions are even partially violated, while our result still provides alternative guarantees under practical assumptions.

## 4 THOUGHTCOMM: Multiagent Communication via Thought

Based on the established theory, we propose a practical framework, THOUGHTCOMM, for multi-agent collaboration in which agents exchange *thoughts* directly. At each communication round $t$, we first encode the agents' model states into a shared latent space that captures their internal thoughts. These latent thoughts are then processed and selectively reintegrated into each agent's context based on the structured relationship between thoughts and agents. This allows each agent to gain a global sense of what others are thinking, and to distinguish which thoughts are shared or agent-specific.

### 4.1 Uncovering the Latent Thoughts

Each agent $A_i$ maintains a model state $H_t^{(i)} \in \mathbb{R}^{n_{h_i}}$ corresponding to the representation of its last generated token immediately before communication round $t$, contextualizing the text summarizing their own response. We concatenate these states from all $n$ agents into a single vector as in Eq. 1.

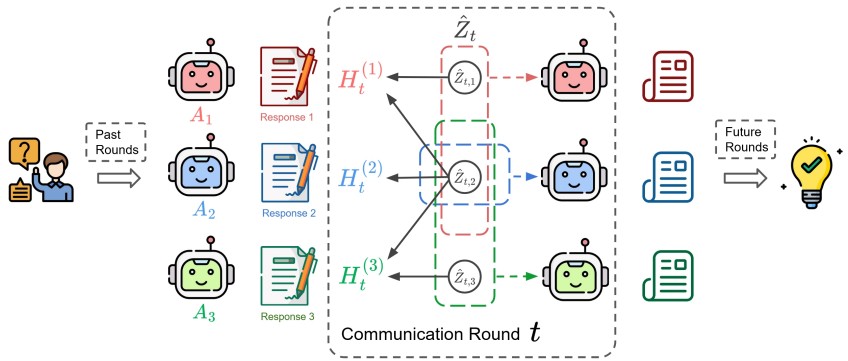

Figure 2: Overview of THOUGHTCOMM. At each communication round $t$, agents encode their model states $H_t^{(i)}$ into a shared latent space via a sparsity-regularized autoencoder, yielding latent thoughts $\hat{Z}_t$. Each dimension $\hat{Z}_{t,j}$ is selectively routed to relevant agents based on the recovered dependency structure, allowing agents to identify both shared and private thoughts for reasoning. The corresponding latent thoughts are then injected into each agent model via prefix adaptation to guide the next response. These updated responses form the input to the next round, enabling multi-agent collaboration beyond purely message exchange.

Then we aim to uncover the hidden process that generate these states from the latent thought of agents. According to the formulation in §2, there exists an underlying process $f$ that generates the agents' responses $H_t$ based on their hidden thoughts $Z_t$, i.e., $H_t = f(Z_t)$.

In the proposed framework, the concatenated state $H_t$ is mapped into a latent space via a *sparsity-regularized autoencoder* with $\ell_1$ regularization on $J_{\hat{f}}$. The resulting latent thoughts $\hat{Z}_t$ are recovered through its encoder $\hat{f}^{-1}$:

$$\hat{Z}_t = \hat{f}^{-1}(H_t) \in \mathbb{R}^{n_z}. \tag{6}$$

The connection between our estimation $\hat{Z}_t$ and the ground-truth latent thoughts $Z_t$ is built by our identifiability theory established in §3. The structure of the latent thought $Z_t$ is governed by the Jacobian $J_f(Z_t) \in \mathbb{R}^{n_h \times n_z}$, whose non-zero pattern $B_{J_f}$ reveals which latent dimensions are influenced by which agents' states. The autoencoder is trained to reconstruct the full state vector:

$$\mathcal{L}_{\text{rec}} = \left\| H_t - \hat{f}(\hat{Z}_t) \right\|_2^2 + \left\| J_{\hat{f}} \right\|_1, \tag{7}$$

ensuring consistency between $H_t$ and its reconstruction via $\hat{Z}_t$, as well as the required sparsity regularization on the Jacobian. This enforces observational equivalence between the estimated and ground-truth processes, which serves as the foundation for identifiability. At test time, we use the trained encoder $\hat{f}^{-1}$ to extract latent thoughts $\hat{Z}_t$ from hidden states $H_t$, and leverage the recovered dependency structure $B_{J_{\hat{f}}}$ to determine which latent dimensions of $\hat{Z}_t$ are relevant for each agent.

### 4.2 Leveraging the Structure of Thoughts

To provide personalized access to latent thoughts, we adopt an agreement-based reweighting strategy. Specifically, for agent $A_i$ at communication round $t$, we first identify the set of latent thoughts $\hat{Z}_{H_t^{(i)}}$ that influence its model state, i.e., $\hat{Z}_{H_t^{(i)}} := \left\{ \hat{Z}_{t,j} \in \hat{Z}_t \mid \exists q \in [i_l, i_h] \text{ such that } B(J_{\hat{f}})_{q,j} \neq 0 \right\}$. These latent thoughts are then partitioned into groups based on their level of agreement across agents, measured by the number of agents whose hidden states in $\hat{H}_t$ depend on each latent dimension in thoughts $\hat{Z}_t$. Formally, for every $\hat{Z}_{t,j} \in \hat{Z}_{H_t^{(i)}}$, we define its agent agreement as:

$$\alpha_j = \sum_{k=1}^{n_a} \mathbb{I}\left( \hat{Z}_{t,j} \in \hat{Z}_{H_t^{(k)}} \right), \tag{8}$$

where $\mathbb{I}(\cdot)$ is the indicator function. Latent thoughts are then grouped by their agreement level $\alpha_j$.

Each group is assigned a distinct weight $w_{\alpha_j}$, reflecting the relevance or generality of these thoughts across agents. The new latent representation for agent $A_i$ is constructed by combining all groups

$$\tilde{Z}_t^{(i)} = \text{concat}_\alpha(w_{\alpha_j} \cdot \hat{Z}_{t,\alpha}^{(i)}), \tag{9}$$

where $\hat{Z}_{t,\alpha}^{(i)}$ denotes the subset of latent variables in $\hat{Z}_{H_t^{(i)}}$ with agreement level $\alpha$, i.e.,

$$\hat{Z}_{t,\alpha}^{(i)} = \left\{ \hat{Z}_{t,j} \in \hat{Z}_{H_t^{(i)}} \;\Big|\; \alpha_j = \alpha \right\}. \tag{10}$$

Intuitively, the recovered dependency structure plays a critical role in shaping how latent thoughts are routed to each agent. After extracting the shared latent space via the sparsity-regularized autoencoder, we apply a structural mask to ensure that each agent only receives the latent dimensions that are relevant to its own internal representation. This filtering directly affects how the injected prefixes are constructed for each agent during the next round of generation. The agreement weights further distinguish different types of relevant thoughts. Although the surface-level messages are broadcast, the actual content used to condition each agent's reasoning is selectively and adaptively constructed in the latent space, reflecting the personalized structure of shared and private thoughts.

### 4.3 Latent Injection via Prefix Adaptation

To seamlessly integrate the recovered latent thoughts into agent behavior, we incorporate them into the generation process via *prefix adaptation*. For each agent $A_i$, we construct a prefix vector from its personalized latent representation $\tilde{Z}_t^{(i)}$ via a learned adapter function:

$$P_t^{(i)} = g(\tilde{Z}_t^{(i)}) \in \mathbb{R}^{m \times d}, \tag{11}$$

where $m$ is the prefix length and $d$ is the embedding dimension. Following Li and Liang [2021], we prepend the resulting prefix $P_t^{(i)}$ to the token embeddings of agent $A_i$ in the next generation step, leveraging the latent thoughts to guide response generation without explicit message passing.

To train the adapter $g$, we inject its output as a prefix and generate a brief continuation (e.g., one sentence), keeping it short to focus on linguistic coherence rather than influencing the actual solution. The few generated tokens are compared against a reference using a semantic similarity loss and a standard regularization term that promotes linguistic fluency:

$$\mathcal{L}_{\text{comm}} = \sum_{i=1}^{n_a} \sum_{t=1}^{T} \left[ \left( 1 - \cos\left( \bar{\phi}(y_{t,i}^{\text{gen}}), \bar{\phi}(y_{t,i}^{\text{ref}}) \right) \right) - \log p(y_{t,i}^{\text{gen}} \mid \text{context}_{t,i}, P_t^{(i)}) \right], \tag{12}$$

where $y_{t,i}^{\text{gen}}$ denotes the tokens generated by agent $A_i$ at round $t$, $y_{t,i}^{\text{ref}}$ is a reference from the model without latent communication, $\text{context}_{t,i}$ denotes the dialogue history or prompt available to agent $A_i$, and $P_t^{(i)}$ is the injected prefix produced by the adapter. $\bar{\phi}(\cdot)$ denotes the mean token embedding. The goal is not to replicate the content of baseline generations, but to ensure that the adapter produces latent modifications whose injected effects remain linguistically natural.

***Remark*** 1. Since the autoencoder is trained only to reconstruct model states, and the adapter is guided simply to avoid producing semantically absurd responses, both components remain largely *task-agnostic* and can be pretrained once and reused. This modular design allows latent communication to be applied across different tasks without retraining, enabling easy integration into multi-agent generation systems with minimal overhead.

## 5 Experiments

In this section, we conduct both synthetic and real-world experiments across various settings. Part of the implementation details are deferred to Appx. C.

### 5.1 Synthetic Evaluation

We begin with synthetic experiments to validate the identifiability of latent thoughts. For the basic setup corresponding to our running example in Fig. 1, we consider two observed variables, $X_A$ and $X_B$, and three latent ones: $Z_A \setminus Z_B$, $Z_B \setminus Z_A$, and $Z_A \cap Z_B$, to evaluate whether shared and private

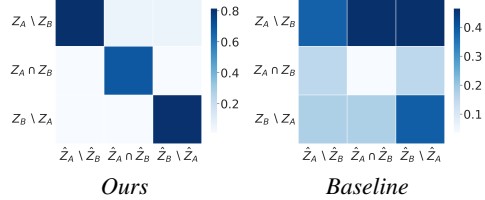

Figure 3: $R^2$ of two models.

latent variables can be correctly recovered. The datasets are generated by a random invertible transformation from multivariate Laplacian variables. We train a sparsity-regularized autoencoder on these datasets and compute the standard $R^2$ score between each part of the estimated and ground-truth latents. A baseline model without sparsity regularization is also included for comparison.

The results are shown in Fig. 3. A higher $R^2$ indicates closer correspondence between the estimated latent variables and the matching ground-truth components, and vice versa. Our model clearly identifies the shared region $Z_A \cap Z_B$ and the private regions $Z_A \setminus Z_B$ and $Z_B \setminus Z_A$, while the baseline fails to disentangle them.

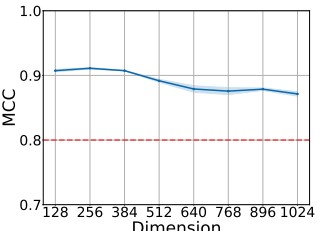

Beyond the basic setup, we evaluate whether incorporating multiple pairs of observed variables in a complex system enables recovery of most latent variables, as considering all pairs of agents reveals exponentially more information than any single pair alone. Following the identifiability literature, we compute the mean correlation

Figure 4: MCC across setups.

coefficient (MCC) between estimated and ground-truth latents across 8 settings, with dimensionality ranging from 124 to 1024 and equal numbers of latent and observed variables. Results are shown in Fig. 4. The *red line* marks the threshold typically considered identifiable when exceeded. Our model consistently recovers most latent variables across all settings, highlighting the global identifiability.

## 5.2 Real-World Evaluation

Recent empirical analyses [Cemri et al., 2025, Hu et al., 2025] reveal that LLM-based multi-agent systems frequently struggle with reasoning tasks, demonstrating only modest improvements over strong single-agent baselines due to coordination inefficiencies and communication bottlenecks – challenges that THOUGHTCOMM is explicitly expected to address. Therefore, we evaluate THOUGHTCOMM on two widely used math reasoning benchmarks, MATH [Hendrycks et al., 2021] and GSM8K [Cobbe et al., 2021] to assess its real-world effectiveness. For the main experiments in this section, we follow Subramaniam et al. [2025] by using three agents engaging in two rounds of debate.

**Baselines.** As the proposed THOUGHTCOMM introduces an additional training stage, the most direct baseline is Multiagent Finetuning [Subramaniam et al., 2025], which is the current state-of-the-art in maximizing multi-agent collaboration through specialized roles and multiple finetuning rounds. We also include single-LLM performance, referred to as "single answer," for comparison. It is worth noting that there are many other multi-agent collaboration workflows; our objective here is to validate the potential of the proposed paradigm rather than exhaustively compare all possible strategies.

**Data pre-processing and evaluation metrics.** Following Subramaniam et al. [2025], we randomly sample 500 examples for fine-tuning the latent communication module, which includes both an autoencoder and an adapter, while reserving another 500 examples for evaluation. We select the more challenging questions for evaluation (e.g., level-3 complexity in MATH [Hendrycks et al., 2021]) when applicable. Generated responses are parsed and evaluated against the ground truths, with *accuracy* measured as the percentage of correctly generated answers. To quantify the reliability of these estimates, we also report the standard deviation for each accuracy score. Beside accuracy, we include a *consensus* score, defined as the proportion of final-round instances where all agents reached a unanimous decision, providing a more direct measure of communication effectiveness.

**Models.** We evaluated both the baseline methods and THOUGHTCOMM on five latest LLMs of varying model sizes, including Llama-3-8B-Instruct [Grattafiori et al., 2024], Phi-4-mini-instruct [Abdin et al., 2024], Qwen-3-0.6B, Qwen-3-1.7B [Yang et al., 2025], as well as the Deepseek-R1-distilled-Llama-8B [Guo et al., 2025].

**Main results.** Table 1 presents the main results, showing that THOUGHTCOMM consistently outperforms baseline methods across both the MATH [Hendrycks et al., 2021] and GSM8K [Cobbe et al., 2021] benchmarks. Within all base models, THOUGHTCOMM demonstrates clear improvements over both single answer and Multiagent Finetuning [Subramaniam et al., 2025]. For instance, on Qwen 3-1.7B, THOUGHTCOMM achieves 93% accuracy on MATH, representing an 17.2% absolute gain over Multiagent Finetuning and a 113.3% relative improvement over the single answer baseline. On average, THOUGHTCOMM achieves 67.23% relative improvement over single answer and 19.06% over the current state-of-the-art. In terms of consensus, THOUGHTCOMM also outperforms all baselines by a clear margin, with its improved consensus directly translating to higher accuracy, indicating superior inter-agent alignment enabled by efficient mind-to-mind communication. These gains are consistently observed across models ranging from 0.6B to 8B parameters, demonstrating the scalability and robustness of the proposed approach across a broad range of model sizes.

Additionally, unlike Multiagent Finetuning [Subramaniam et al., 2025], which requires finetuning the entire LLM and thus incurs substantial overhead, THOUGHTCOMM only trains a lightweight

Table 1: Evaluation results on MATH [Hendrycks et al., 2021] and GSM8K [Cobbe et al., 2021] for various methods with five different LLMs. Bold numbers indicate the best performance.

| Base Model | Methods | MATH | | GSM8K | |
|---|---|---|---|---|---|
| | | Accuracy (%) | Consensus (%) | Accuracy (%) | Consensus (%) |
| Qwen 3-0.6B | Single Answer | 45.80 ± 2.23 | N/A | 58.20 ± 2.21 | N/A |
| | Multiagent Finetuning | 71.20 ± 2.03 | 90.07 | 70.80 ± 2.03 | 86.40 |
| | THOUGHTCOMM | **85.00 ± 1.60** | **91.20** | **75.80 ± 1.92** | **89.27** |
| Qwen 3-1.7B | Single Answer | 43.60 ± 2.22 | N/A | 67.40 ± 2.10 | N/A |
| | Multiagent Finetuning | 75.80 ± 1.92 | 95.80 | 84.20 ± 1.63 | 96.73 |
| | THOUGHTCOMM | **93.00 ± 1.14** | **95.93** | **85.00 ± 1.60** | **97.87** |
| Phi-4-mini-instruct (3.84B) | Single Answer | 63.80 ± 2.15 | N/A | 81.60 ± 1.73 | N/A |
| | Multiagent Finetuning | 60.20 ± 2.19 | 78.89 | 82.16 ± 1.71 | 91.24 |
| | THOUGHTCOMM | **74.60 ± 1.95** | **84.73** | **84.20 ± 1.63** | **94.73** |
| LLaMA 3-8B-Instruct | Single Answer | 36.20 ± 2.15 | N/A | 60.80 ± 2.18 | N/A |
| | Multiagent Finetuning | 39.68 ± 2.19 | 68.97 | **69.20 ± 2.06** | 80.20 |
| | THOUGHTCOMM | **45.60 ± 2.23** | **74.67** | 68.40 ± 2.08 | **84.87** |
| DeepSeek-R1-Distill-Llama-8B | Single Answer | 42.60 ± 2.21 | N/A | 65.60 ± 2.12 | N/A |
| | Multiagent Finetuning | 72.40 ± 2.00 | **82.87** | 76.80 ± 1.89 | 83.13 |
| | THOUGHTCOMM | **82.80 ± 1.69** | 80.72 | **80.80 ± 1.76** | **88.13** |

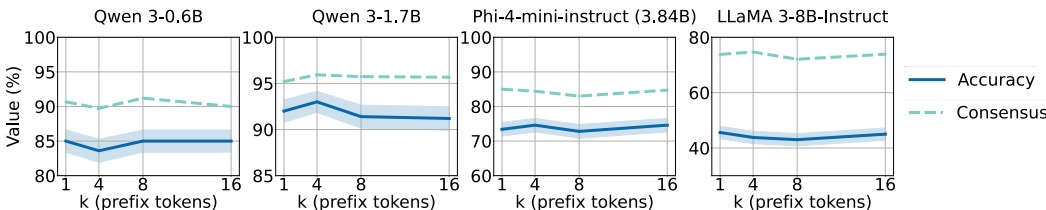

Figure 5: Two-agent THOUGHTCOMM with accuracy (solid) and consensus (dashed) performance on MATH [Hendrycks et al., 2021] as prefix length increases from 1 to 16.

autoencoder and adapter, whose computational cost depends only on the LLM's embedding dimension rather than the parameter count. This results in fundamentally smaller and model-agnostic training overhead, enabling efficient and scalable deployment even for very large LLMs. For instance, both Llama-3-70B and 405B share a 16,384 embedding dimension; thus, THOUGHTCOMM's overhead remains unchanged when moving from 70B to 405B, whereas Multiagent Finetuning [Subramaniam et al., 2025] would require substantially more training cost at each scale. Overall, these results validate both the efficiency and effectiveness of the proposed THOUGHTCOMM, supporting the theoretical predictions of enhanced coordination and cognitive alignment in multi-agent LLMs.

## 5.3 Scaling the Number of Debate Rounds

We further investigate how the number of debate rounds impacts multi-agent performance, as more rounds may introduce redundant or confusing information that can degrade results. With two agents, we vary the number of rounds from 2 to 6 and evaluate on the MATH [Hendrycks et al., 2021] benchmark using LLaMA-3-8B-Instruct [Grattafiori et al., 2024], following the setup in §5.2. As shown in Fig. 6, Multiagent

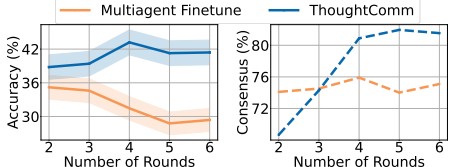

Figure 6: Multi-agent performance as the number of debate rounds increases.

Finetuning suffers a drop in accuracy with more rounds, while consensus slightly increases and maintains. In contrast, THOUGHTCOMM achieves simultaneous gains in both accuracy and consensus, demonstrating robustness to redundancy and noise by consistently identifying true latent *thoughts*.

## 5.4 Varying the Prefix Lengths

As discussed in §4.3, the prefix length $m$ determines how many thought vectors are injected into agent context. A key question is whether THOUGHTCOMM remains robust as $m$ grows, or if excessive prefixes introduce redundant or irrelevant information that degrades performance. To answer these questions, we sweep the prefix length $m \in \{1, 4, 8, 16\}$ across four models with different parameter sizes (Llama-3-8B-Instruct [Grattafiori et al., 2024], Phi-4-mini-instruct [Abdin et al., 2024], Qwen-3

0.6B, and Qwen-3 1.7B [Yang et al., 2025]) on the MATH [Hendrycks et al., 2021] benchmark, using the same 500/500 train/test split from §5.2. As shown in Fig. 5, both accuracy and consensus stay remarkably stable for all four models, with performance fluctuations under five percent even as $m$ increases sixteen-fold. These results demonstrate a clear robustness advantage of THOUGHTCOMM by delivering reliable gains without requiring precise tuning of the prefix length, dramatically reducing hyperparameter overhead in practice. Moreover, achieving near-optimal performance with a single injected vector highlights the efficiency of our thought-communication mechanism. While both token and prefix embeddings have the same dimensionality (e.g., 1024), a token embedding is tied to a single vocabulary item and typically encodes the semantics of just that one discrete token, often lying on a lower-dimensional subspace. In contrast, a prefix embedding is a free parameter optimized to encode many continuous latent thoughts, leveraging the full capacity of the embedding space.

## 6 Related Works

**Multiagent LLMs communication.** LLM-based multi-agent systems (MAS) have become a compelling strategy for advancing beyond the limitations of single LLMs [Li et al., 2023, Wu et al., 2023, Hong et al., 2023, Guo et al., 2024, Tran et al., 2025]. Specifically, multi-agent debate [Du et al., 2023, Pham et al., 2023, Liang et al., 2023], which mimics human collaborative reasoning, has shown particular promise by amplifying reasoning through collective, diverse exchanges. One of the most central factors that determines MAS effectiveness is the communication paradigm between agents [Li et al., 2024, Cemri et al., 2025]. Extensive research has sought to improve this paradigm, exploring various directions such as improving communication efficiency [Zhang et al., 2024a, Wang et al., 2025b, Zeng et al., 2025], enabling more flexible topologies and workflows [Khattab et al., 2023, Zhang et al., 2024b, Liu et al., 2024, Wu et al., 2024, Wang et al., 2024, 2025a], mitigating error propagation [Wang et al., 2023, Yoffe et al., 2024], shifting from turn-based, full-response discussion to token-level collaboration [Bian et al., 2025, Chakraborty et al., 2025], and moving beyond text tokens to token embeddings [Pham et al., 2023]. However, all these approaches fundamentally rely on the exchange of natural language, either through text tokens or their embeddings, thus inheriting the constraints of human-style communication. In contrast, THOUGHTCOMM pioneers a new communication paradigm by extracting and uncovering the underlying *latent thoughts* beneath surface-level language tokens and embeddings, enabling a more direct and expressive form of MAS communication and collaboration.

**Identifiability of latent variable models.** Classical identifiability results in latent variable models largely focus on linear settings, offering strong guarantees through factor analysis, structural equations, and ICA [Reiersøl, 1950, Lawley and Maxwell, 1962, Aigner et al., 1984, Comon, 1994, Bekker and ten Berge, 1997, Bishop, 1998]. To relax linearity, previous work introduces auxiliary variables [Hyvärinen and Morioka, 2016, Hyvärinen et al., 2019, Yao et al., 2021, Hälvä et al., 2021, Lachapelle et al., 2022, Song et al., 2024, Li et al., 2025a], structural constraints on the mixing function [Taleb and Jutten, 1999, Moran et al., 2021, Kivva et al., 2022, Zheng et al., 2022, Buchholz et al., 2022, Zheng et al., 2025], or the synergy of both [Zheng and Zhang, 2023, Li et al., 2025b]. Causal representation learning often depends on interventions [von Kügelgen et al., 2023, Jiang and Aragam, 2023, Jin and Syrgkanis, 2023, Zhang et al., 2024c] or counterfactual views [von Kügelgen et al., 2021, Brehmer et al., 2022]. These approaches typically require parametric assumptions or external signals. With a weaker goal of identifying only shared and private thoughts and their structures across agents, our framework can be applied in the general nonparametric setting without such aids.

## 7 Conclusion

To enable LLM agents to communicate through thoughts, we formulate multi-agent communication as a latent variable model to explore agents' minds. We establish identifiability results under general conditions to ensure reliable recovery of latent thoughts and structures, and propose a new framework, THOUGHTCOMM, for effective collaboration via thought. While this introduces a new direction, certain *limitations* remain. Our experiments focus on using model states as observed variables, which may not be feasible for closed-source models. A promising alternative is to replace them with context-aware embeddings of the observational data and recover latent thoughts from those. The observational data need not be textual and can span any modality, extending the framework beyond LLMs. Although we have not explored this empirically, as generating embeddings suitable for summarization is a separate topic, the theory and framework can accommodate this extension directly. We hope this work sheds light on the hidden world beneath observation, as many challenges remain unsolvable through surface-level observation, regardless of scale in data or compute.

## Acknowledgment

The authors would like to thank the anonymous reviewers and AC for helpful comments and suggestions during the reviewing process. The authors would also like to acknowledge the support from NSF Award No. 2229881, AI Institute for Societal Decision Making (AI-SDM), the National Institutes of Health (NIH) under Contract R01HL159805, and grants from Quris AI, Florin Court Capital, MBZUAI-WIS Joint Program, and the Al Deira Causal Education project.

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

# Appendix:
## *Thought Communication in Multiagent Collaboration*

## Table of Contents

## A  Proofs

### A.1  Proof of Theorem 1

**Theorem 1** (Identifying the shared thoughts). *Suppose that for each $i \in [n_x]$, there exist points where the Jacobians $J_f(Z_t)_{i,\cdot}$ span the support subspace $\mathcal{S}_{J_{f_{i,\cdot}}}$, and that $(J_f(Z_t)\mathcal{M})_{i,\cdot} \in \mathcal{S}_{J_{\hat{f}_{i,\cdot}}}$ at those points. If $H_t \overset{d}{=} \hat{f}(\hat{Z}_t)$ for a model $(\hat{f}, \hat{Z}_t)$ following §2 with $\ell_0$ regularization on $J_{\hat{f}}$, then for any pair of agents $A_i$ and $A_j$ at round t, there exists a permutation $\pi$ over $[n_z]$ such that $\frac{\partial Z_i}{\partial \hat{Z}_{\pi(j)}} = 0$ for any $Z_i \in Z_{H_t^{(i)}} \cap Z_{H_t^{(j)}}$ and any $Z_j \in (Z_{H_t^{(i)}} \cup Z_{H_t^{(j)}}) \setminus (Z_{H_t^{(i)}} \cap Z_{H_t^{(j)}})$.*

*Proof.* Because $H_t \overset{d}{=} \hat{f}(\hat{Z}_t)$ and $H_t \overset{d}{=} f(Z_t)$, we have

$$p(\hat{f}(\hat{Z}_t)) = p(f(Z_t)). \tag{13}$$

According to the change-of-variable formula, there exists $h = \hat{f}^{-1} \circ f \colon Z_t \to \hat{Z}_t$ s.t. $\hat{Z}_t = h(Z_t)$. Taking the derivatives of both sides w.r.t. $Z_t$ yields

$$J_f(Z_t) = J_{\hat{f}}(\hat{Z}_t) J_h(Z_t), \tag{14}$$

which is equivalent to

$$J_{\hat{f}}(\hat{Z}_t) = J_f(Z_t) J_h^{-1}(Z_t). \tag{15}$$

The inverse Jacobian $J_h^{-1}(Z_t)$ exists since both $f$ and $\hat{f}$ are invertible, implying that $h = \hat{f}^{-1} \circ f$ is itself invertible.

Since for each $i \in [n_z]$, there exist points where the Jacobian $J_f(Z_t)_{i,\cdot}$ spans its support subspace $(\mathcal{S}_{J_f})_{i,\cdot}$, we can express any vector in that subspace with a linear combination of these vectors. Therefore, for any $j \in [n_z]$ where $B(J_f)_{i,j} \neq 0$, we have

$$M_{j,\cdot} = e_j M, \tag{16}$$

where $M$ is a constant matrix with the same nonzero pattern as $J_h^{-1}(Z_t)$, and we construct a one-hot vector $e_j \in \mathcal{S}_{J_{f_{i,\cdot}}}$ with $\alpha_k$ as coefficients of that linear combination:

$$e_j := \sum_{k \in B_i} \alpha_k (J_f(z^{(k)}))_{i,\cdot}, \tag{17}$$

where $B_i$ denotes the set of points spanning the subspace. Thus we have

$$M_{j,\cdot} = \sum_{k \in B_i} \alpha_k (J_f(z^{(k)}))_{i,\cdot} M. \tag{18}$$

According to the assumption, we have

$$(J_f(z^{(k)}))_{i,\cdot} M = (J_f(Z_t)M)_{i,\cdot} \in \mathcal{S}_{J_{\hat{f}_{i,\cdot}}}. \tag{19}$$

Therefore, for any $j \in [n_z]$ where $B(J_f)_{i,j} \neq 0$ there is

$$M_{j,\cdot} \in \mathcal{S}_{J_{\hat{f}_{i,\cdot}}}. \tag{20}$$

Construct a bipartite graph

$$G = (R, C, E), \quad R = C = [n_z], \quad (j,k) \in E \iff M_{j,k} \neq 0.$$

Since $M$ is invertible, its rows are linearly independent, giving

$$|N(S)| \geq |S| \quad \forall S \subseteq R, \tag{21}$$

where $N(S)$ is the neighborhood of $S$. Hall's marriage theorem then yields a permutation $\pi \in S_{n_z}$ with

$$J_h^{-1}(Z_t)_{j,\pi(j)} \neq 0, \quad \forall j \in [n_z]. \tag{22}$$

According to Eq. (20), this further implies that, for any $j \in [n_z]$ where $B(J_f)_{i,j} \neq 0$, there is

$$(i, \pi(j)) \in i \times M_{j,\cdot} \subset S_{J_{\hat{f}}} \tag{23}$$

Hence

$$(J_f(Z_t))_{i,j} \neq 0 \implies (J_{\hat{f}}(\hat{Z}_t))_{i,\pi(j)} \neq 0. \tag{24}$$

Given additionally the $\ell_0$ regularization on $J_{\hat{f}}$:

$$\|(J_{\hat{f}})_{i,\cdot}\|_0 \leq \|(J_f)_{i,\cdot}\|_0, \quad \forall i \in [n_z]. \tag{25}$$

Together with (24), this gives the equivalence

$$(J_f(Z_t))_{i,j} \neq 0 \iff (J_{\hat{f}}(\hat{Z}_t))_{i,\pi(j)} \neq 0, \quad \forall i, j \in [n_z], \tag{26}$$

For any $Z_i \in Z_{H_t^{(i)}} \cap Z_{H_t^{(j)}}$ and any $Z_j \in (Z_{H_t^{(i)}} \cup Z_{H_t^{(j)}}) \setminus (Z_{H_t^{(i)}} \cap Z_{H_t^{(j)}})$, there is

$$B(J_f)_{H_t^{(i)},i} \neq 0. \tag{27}$$

Based on Eq. (20), there is

$$M_{i,\cdot} \in \mathcal{S}_{J_{\hat{f}_{H_t^{(i)},\cdot}}}, \tag{28}$$

where we use $\mathcal{S}_{J_{\hat{f}_{H_t^{(i)},\cdot}}}$ to index multiple rows corresponding to $H_t^{(i)}$ at once. This is for notational brevity and will also be applied later. We also have

$$B(J_f)_{H_t^{(j)},i} \neq 0, \tag{29}$$

where we slightly abuse the notation to indicate that not all entries at the specified indices are zero. This convention is adopted throughout, though we only make it explicit here.

Similarly, there is also

$$M_{i,\cdot} \in \mathcal{S}_{J_{\hat{f}_{H_t^{(j)},\cdot}}}. \tag{30}$$

Suppose for contradiction that, for any $Z_j \in (Z_{H_t^{(i)}} \cup Z_{H_t^{(j)}}) \setminus (Z_{H_t^{(i)}} \cap Z_{H_t^{(j)}})$, there is

$$M_{i,\pi(j)} \neq 0. \tag{31}$$

Then, according to Eq. (28), there is

$$B(J_{\hat{f}})_{H_t^{(i)},\pi(j)} \neq 0. \tag{32}$$

This implies the follows according to Eq. (26):

$$B(J_f)_{H_t^{(i)},j} \neq 0. \tag{33}$$

Similarly, according to Eq. (30), there is

$$B(J_{\hat{f}})_{H_t^{(j)},\pi(j)} \neq 0. \tag{34}$$

This implies the follows according to Eq. (26):

$$B(J_f)_{H_t^{(j)},j} \neq 0. \tag{35}$$

Thus, there must be

$$Z_j \in Z_{H_t^{(i)}} \cap Z_{H_t^{(j)}}, \tag{36}$$

which contradicts

$$Z_j \in (Z_{H_t^{(i)}} \cup Z_{H_t^{(j)}}) \setminus (Z_{H_t^{(i)}} \cap Z_{H_t^{(j)}}). \tag{37}$$

Therefore, there must be

$$M_{i,\pi(j)} = 0, \tag{38}$$

which implies $\frac{\partial Z_i}{\partial \hat{Z}_{\pi(j)}} = 0$. □

## A.2 Proof of Theorem 2

**Theorem 2** (Identifying the private thoughts). *Suppose the assumption in Thm. 1 holds. If $H_t \overset{d}{=} \hat{f}(\hat{Z}_t)$ for a model $(\hat{f}, \hat{Z}_t)$ following §2 with $\ell_0$ regularization on $J_{\hat{f}}$, then for any pair of agents $A_i$ and $A_j$ at round $t$, there exists a permutation $\pi$ over $[n_z]$ such that $\frac{\partial Z_i}{\partial \hat{Z}_{\pi(j)}} = 0$ for any $Z_i \in Z_{H_t^{(i)}} \setminus Z_{H_t^{(j)}}$ and any $Z_j \in Z_{H_t^{(j)}}$.*

*Proof.* Part of the derivations has been provided in proofs of Theorem 1, and we include it for completeness. Because $H_t \overset{d}{=} \hat{f}(\hat{Z}_t)$ and $H_t \overset{d}{=} f(Z_t)$, we have

$$p(\hat{f}(\hat{Z}_t)) = p(f(Z_t)). \tag{39}$$

According to the change-of-variable formula, there exists $h = \hat{f}^{-1} \circ f \colon Z_t \to \hat{Z}_t$ s.t. $\hat{Z}_t = h(Z_t)$. Taking the derivatives of both sides w.r.t. $Z_t$ yields

$$J_f(Z_t) = J_{\hat{f}}(\hat{Z}_t) J_h(Z_t), \tag{40}$$

which is equivalent to

$$J_{\hat{f}}(\hat{Z}_t) = J_f(Z_t) J_h^{-1}(Z_t). \tag{41}$$

The inverse Jacobian $J_h^{-1}(Z_t)$ exists since both $f$ and $\hat{f}$ are invertible, implying that $h = \hat{f}^{-1} \circ f$ is itself invertible.

Since for each $i \in [n_z]$, there exist points where the Jacobian $J_f(Z_t)_{i,\cdot}$ spans its support subspace $(S_{J_f})_{i,\cdot}$, we can express any vector in that subspace with a linear combination of these vectors. Therefore, for any $j \in B(J_f)_{i,\cdot}$, we have

$$M_{j,\cdot} = e_j M, \tag{42}$$

where $M$ is a constant matrix with the same nonzero pattern as $J_h^{-1}(Z_t)$, and we construct a one-hot vector $e_j \in S_{J_{f_{i,\cdot}}}$ with $\alpha_k$ as coefficients of that linear combination:

$$e_j := \sum_{k \in B_i} \alpha_k (J_f(z^{(k)}))_{i,\cdot}, \tag{43}$$

where $B_i$ denotes the set of points spanning the subspace. Thus we have

$$M_{j,\cdot} = \sum_{k \in B_i} \alpha_k (J_f(z^{(k)}))_{i,\cdot} M. \tag{44}$$

According to the assumption, we have

$$(J_f(z^{(k)}))_{i,\cdot}M = (J_f(Z_t)M)_{i,\cdot} \in \mathcal{S}_{J_{f_{i,\cdot}}}. \tag{45}$$

Therefore, for any $j \in B(J_f)_{i,\cdot}$, there is

$$M_{j,\cdot} \in \mathcal{S}_{J_{\hat{f}_{i,\cdot}}}. \tag{46}$$

Construct a bipartite graph

$$G = (R, C, E), \quad R = C = [n_z], \quad (j,k) \in E \iff M_{j,k} \neq 0.$$

Since $M$ is invertible, its rows are linearly independent, giving

$$|N(S)| \geq |S| \quad \forall S \subseteq R, \tag{47}$$

where $N(S)$ is the neighborhood of $S$. Hall's marriage theorem then yields a permutation $\pi \in S_{n_z}$ with

$$J_h^{-1}(Z_t)_{j,\pi(j)} \neq 0, \quad \forall j \in [n_z]. \tag{48}$$

According to Eq. (46), this further implies that, for any $j \in [n_z]$ where $B(J_f)_{i,j} \neq 0$, there is

$$(i, \pi(j)) \in i \times M_{j,\cdot} \subset S_{J_{\hat{f}}} \tag{49}$$

Hence

$$(J_f)_{i,j} \neq 0 \implies (J_{\hat{f}})_{i,\pi(j)} \neq 0. \tag{50}$$

Given additionally the $\ell_0$ regularization on $J_{\hat{f}}$:

$$\|(J_{\hat{f}})_{i,\cdot}\|_0 \leq \|(J_f)_{i,\cdot}\|_0, \quad \forall i \in [n_z]. \tag{51}$$

Together with Eq. (50), this gives the equivalence

$$(J_f(Z_t))_{i,j} \neq 0 \iff (J_{\hat{f}}(\hat{Z}_t))_{i,\pi(j)} \neq 0, \quad \forall i,j \in [n_z], \tag{52}$$

Consider the case where $Z_{i'} \in Z_{H_t^{(i)}} \cap Z_{H_t^{(j)}}$ and $Z_{j'} \in Z_t \setminus (Z_{H_t^{(i)}} \cap Z_{H_t^{(j)}})$. Based on Eq. (46), there is

$$M_{i',\cdot} \in \mathcal{S}_{J_{\hat{f}_{H_t^{(i)},\cdot}}}. \tag{53}$$

Suppose

$$M_{i',\pi(j')} \neq 0. \tag{54}$$

Then we have

$$B(J_{\hat{f}})_{H_t^{(i)},\pi(j')} \neq 0, \tag{55}$$

which implies

$$B(J_f)_{H_t^{(i)},j'} \neq 0. \tag{56}$$

At the same time, since $Z_{i'} \in Z_{H_t^{(j)}}$, there is

$$M_{i',\cdot} \in \mathcal{S}_{J_{\hat{f}_{H_t^{(j)},\cdot}}}. \tag{57}$$

Since we suppose $M_{i',\pi(j')} \neq 0$, it follows that

$$B(J_{\hat{f}})_{H_t^{(j)},\pi(j')} \neq 0, \tag{58}$$

which implies

$$B(J_f)_{H_t^{(j)},j'} \neq 0. \tag{59}$$

Clearly, Eqs. (56) and (59) together contradict $Z_{j'}' \in Z_t \setminus (Z_{H_t^{(i)}} \cap Z_{H_t^{(j)}})$. Thus, there must be

$$M_{i',\pi(j')} = 0. \tag{60}$$

For any $Z_i \in Z_{H_t^{(i)}} \setminus Z_{H_t^{(j)}}$ and any $Z_j \in Z_{H_t^{(j)}}$, we first consider $Z_j \in Z_{H_t^{(j)}} \cap Z_{H_t^{(i)}}$. Since $Z_{H_t^{(j)}} \cap Z_{H_t^{(i)}}$ does not intersect with $Z_{H_t^{(i)}} \setminus Z_{H_t^{(j)}}$, $Z_j$ is not a function of $Z_i$. According to the

invertibility and Eq. (60), $Z_{H_t^{(j)}} \cap Z_{H_t^{(i)}}$ can only be an invertible function of $\sigma(\hat{Z}_{H_t^{(j)}} \cap \hat{Z}_{H_t^{(i)}})$, where $\sigma$ denotes the permutation function corresponding to the permutation $\pi$. Further given that $Z_j$ is not a function of $Z_i$ and $Z_i \in Z_{H_t^{(i)}} \setminus Z_{H_t^{(j)}}$, $Z_j$ is also not a function of any variable in $\sigma(\hat{Z}_{H_t^{(j)}} \cap \hat{Z}_{H_t^{(i)}})$, i.e.,

$$M_{i,\pi(j)} = 0. \tag{61}$$

We then consider the other case where $Z_j \in Z_{H_t^{(j)}} \setminus Z_{H_t^{(i)}}$. There is

$$B(J_f)_{H_t^{(i)},i} \neq 0. \tag{62}$$

It implies that

$$M_{i,\cdot} \in \mathcal{S}_{J_{\hat{f}_{H_t^{(i)},\cdot}}}. \tag{63}$$

For $Z_j \in Z_{H_t^{(j)}} \setminus Z_{H_t^{(i)}}$, suppose

$$M_{i,\pi(j)} \neq 0. \tag{64}$$

Then there is

$$B(J_{\hat{f}})_{H_t^{(i)},\pi(j)} \neq 0. \tag{65}$$

Which is equivalent to

$$B(J_f)_{H_t^{(i)},j} \neq 0. \tag{66}$$

This is a contradiction since $Z_j \in Z_{H_t^{(j)}} \setminus Z_{H_t^{(i)}}$.

Therefore, we have

$$M_{i,\pi(j)} = 0. \tag{67}$$

Considering both cases, we prove that $\frac{\partial Z_i}{\partial \hat{Z}_{\pi(j)}} = 0$ for any $Z_i \in Z_{H_t^{(i)}} \setminus Z_{H_t^{(j)}}$ and any $Z_j \in Z_{H_t^{(j)}}$. $\qquad \square$

## A.3  Proof of Theorem 3

**Theorem 3** (Identifying the structure of thoughts). *Suppose the assumption in Thm. 1 holds. If $H_t \overset{d}{=} \hat{f}(\hat{Z}_t)$ for a model $(\hat{f}, \hat{Z}_t)$ following §2 with $\ell_0$ regularization on $J_{\hat{f}}$, then the nonzero pattern $B(J_f)$ is identifiable up to relabelling, i.e., $B(J_{\hat{f}}) = B(J_f)P$ for a permutation matrix $P$.*

*Proof.* Part of the derivations has been provided in proofs of Theorems 1 and 2, and we include it for completeness. Because $H_t \overset{d}{=} \hat{f}(\hat{Z}_t)$ and $H_t \overset{d}{=} f(Z_t)$, we have

$$p(\hat{f}(\hat{Z}_t)) = p(f(Z_t)). \tag{68}$$

According to the change-of-variable formula, there exists $h = \hat{f}^{-1} \circ f \colon Z_t \to \hat{Z}_t$ s.t. $\hat{Z}_t = h(Z_t)$. Taking the derivatives of both sides w.r.t. $Z_t$ yields

$$J_f(Z_t) = J_{\hat{f}}(\hat{Z}_t)J_h(Z_t), \tag{69}$$

which is equivalent to

$$J_{\hat{f}}(\hat{Z}_t) = J_f(Z_t)J_h^{-1}(Z_t). \tag{70}$$

The inverse Jacobian $J_h^{-1}(Z_t)$ exists since both $f$ and $\hat{f}$ are invertible, implying that $h = \hat{f}^{-1} \circ f$ is itself invertible.

Since for each $i \in [n_z]$, there exist points where the Jacobian $J_f(Z_t)_{i,\cdot}$ spans its support subspace $(\mathcal{S}_{J_f})_{i,\cdot}$, we can express any vector in that subspace with a linear combination of these vectors. Therefore, for any $j \in B(J_f)_{i,\cdot}$, we have

$$M_{j,\cdot} = e_j M, \tag{71}$$

where $M$ is a constant matrix with the same nonzero pattern as $J_h^{-1}(Z_t)$, and we construct a one-hot vector $e_j \in \mathcal{S}_{J_{f_{i,\cdot}}}$ with $\alpha_k$ as coefficients of that linear combination:

$$e_j := \sum_{k \in B_i} \alpha_k (J_f(z^{(k)}))_{i,\cdot}. \tag{72}$$

Thus we have

$$M_{j,\cdot} = \sum_{k \in B_i} \alpha_k (J_f(z^{(k)}))_{i,\cdot} M. \tag{73}$$

According to the assumption, we have

$$(J_f(z^{(k)}))_{i,\cdot} M = (J_f(Z_t)M)_{i,\cdot} \in \mathcal{S}_{J_{\hat{f}_{i,\cdot}}}. \tag{74}$$

Therefore, for any $j \in B(J_f)_{i,\cdot}$, there is

$$M_{j,\cdot} \in \mathcal{S}_{J_{\hat{f}_{i,\cdot}}}. \tag{75}$$

Construct a bipartite graph

$$G = (R, C, E), \quad R = C = [n_z], \quad (j,k) \in E \iff M_{j,k} \neq 0.$$

Since $M$ is invertible, its rows are linearly independent, giving

$$|N(S)| \geq |S| \quad \forall S \subseteq R,$$

where $N(S)$ is the neighborhood of $S$. Hall's marriage theorem then yields a permutation $\pi \in S_{n_z}$ with

$$J_h^{-1}(Z_t)_{j,\pi(j)} \neq 0, \quad \forall j \in [n_z]. \tag{76}$$

According to Eq. (75), this further implies that, for any $j \in [n_z]$ where $B(J_f)_{i,j} \neq 0$, there is

$$(i, \pi(j)) \in i \times M_{j,\cdot} \subset S_{J_{\hat{f}}} \tag{77}$$

Hence

$$(J_f)_{i,j} \neq 0 \implies (J_{\hat{f}})_{i,\pi(j)} \neq 0. \tag{78}$$

Given additionally the $\ell_0$ regularization on $J_{\hat{f}}$:

$$\|(J_{\hat{f}})_{i,\cdot}\|_0 \leq \|(J_f)_{i,\cdot}\|_0, \quad \forall i \in [n_z]. \tag{79}$$

Together with (78), this gives the equivalence

$$(J_f(Z_t))_{i,j} \neq 0 \iff (J_{\hat{f}}(\hat{Z}_t))_{i,\pi(j)} \neq 0, \quad \forall i,j \in [n_z]. \tag{80}$$

This implies the equation that

$$B(J_{\hat{f}}) = B(J_f)P, \tag{81}$$

where $P$ is a permutation matrix. □

## B  Supplementary Discussion

**Alternative to model states.** Our main framework assumes access to the model states $H_t$ of each agent before communication. These states provide a rich representation of the agent's processing of context and are used as inputs to our autoencoder for recovering latent thoughts. However, such internal states may be inaccessible in many practical settings, particularly when using closed-source or API-restricted models.

In these cases, a viable alternative is to replace the model state $H_t^{(i)}$ of each agent with a compact embedding extracted from its textual response. Specifically, one can apply a context-aware embedding model to summarize the agent's generated text into a fixed-size vector, which is then treated as a proxy for the unavailable model state.

Crucially, this embedding does not need to preserve any structure among agents, nor does it need to reflect the agent's intent. Its only requirement is to provide a compressed summary of the textual content at the linguistic level. Examples of such embedding methods include those from models like BERT or RoBERTa, pooled sentence embeddings from Sentence-BERT [Reimers and Gurevych, 2019], or output vectors from instruction-tuned embedding APIs. These methods are designed to produce compact, semantically meaningful vectors that summarize the surface content of a given text.

Once such an embedding is obtained for each agent, the rest of the framework remains unchanged. The collection of response embeddings is treated as a surrogate for $H_t$ and passed through the sparsity-regularized autoencoder to recover latent thoughts $\hat{Z}_t$. From that point on, latent communication

proceeds identically: inferring shared/private thoughts, routing them based on recovered structure, and injecting them into agents via prefix adaptation.

This replacement provides a drop-in mechanism to support latent communication in scenarios where model internals are inaccessible, enabling broader applicability of the framework across both open- and closed-source agents. Naturally, one may choose suitable encoders for other modalities to extend the framework beyond LLMs.

## C    Experimental Details and Additional Results

### C.1    Implementation Details

For experiments conducted in §5, we set the prefix token count for our method to 1. For baseline comparisons, we utilize the original code released by the authors[†]. All experiments are conducted on a single compute node with 8 NVIDIA H100 GPUs.

### C.2    Additional Results on Scaling Debate Rounds

In §5.3, we compare the performance of Multiagent Finetune [Subramaniam et al., 2025] and THOUGHTCOMM as the number of debate rounds increases from 2 to 6 based on Llama-3-8B-Instruct [Grattafiori et al., 2024]. Here, we further extend the analysis to an additional model, Qwen-3-1.7B [Yang et al., 2025], demonstrating that THOUGHTCOMM remains robust and is not adversely affected by increased redundancy caused by increased numbers of debate rounds.

As shown in Fig. 7, we observe that the accuracy and consensus of THOUGHTCOMM remain stable or even improve as the number of debate rounds increases up to 6. In contrast, the performance of Multiagent Finetune [Subramaniam et al., 2025] declines noticeably as rounds increase beyond 4, particularly in the accuracy metric. This further supports our claim that THOUGHTCOMM is robust to the accumulation of redundant or noisy information introduced by additional communication rounds.

It is important to note, however, that high consensus among agents does not always imply high task accuracy. This phenomenon is particularly evident in the Qwen-3-1.7B [Yang et al., 2025] results for Multiagent Finetune [Subramaniam et al., 2025], where consensus steadily increases as the number of debate rounds grows—from 2 to 6, while the corresponding accuracy remains stagnant or even degrades. This decoupling suggests that agents can converge on a common answer even when that answer is incorrect, leading to a failure mode in which additional communication drives premature agreement rather than genuine reasoning improvements.

In contrast, THOUGHTCOMM not only increases consensus but also aligns higher agreement with improved accuracy. We also highlight that the gap between THOUGHTCOMM and the baseline widens at higher round counts. These results underscore the importance of structure-aware latent communication in preventing unproductive conformity and fostering truly collaborative reasoning in multi-agent LLM systems. Taken together, these findings confirm the scalability of our approach: THOUGHTCOMM enables multi-agent systems to leverage more communication rounds for improved reasoning without incurring the degradation commonly observed in prior debate-style frameworks.

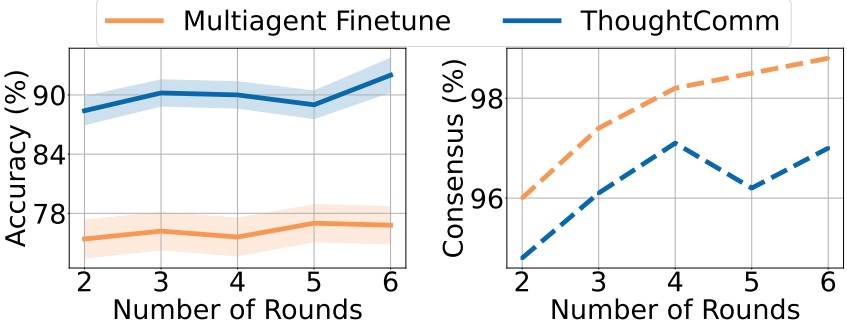

Figure 7:  Additional results of multi-agent performance on Qwen-3-1.7B [Yang et al., 2025] as the number of debate rounds increases.

---

[†] https://github.com/vsubramaniam851/multiagent-ft/tree/main

## C.3 Additional Results on Varying Latent Dimensions

We investigate how varying the latent dimensionality affects performance on the MATH dataset. In these experiments, the setup involves two agents, two rounds, and a single prefix token used for communication. Results are shown for both Llama-3-8B-Instruct and Qwen-3-1.7B models.

As shown in Fig. 8 and Fig. 9, accuracy consistently improves as the latent dimension increases up to 512, after which the gains saturate. This suggests that while higher-capacity latent spaces facilitate richer communication between agents, overly large latent dimensions yield diminishing returns, likely due to redundancy in the learned representations.

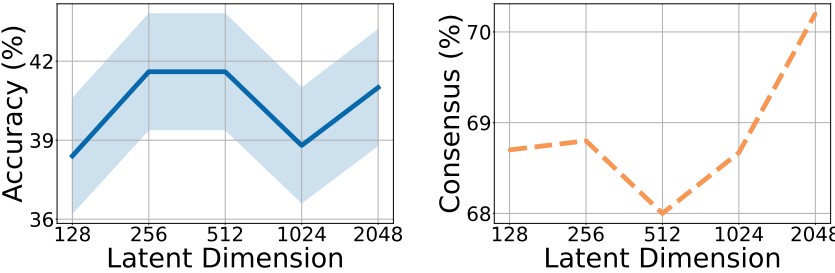

Figure 8: Effect of varying latent dimensionality on MATH for Llama-3-8B-Instruct [Grattafiori et al., 2024]. Accuracy improves with increased latent capacity, stabilizing beyond 1024 dimensions.

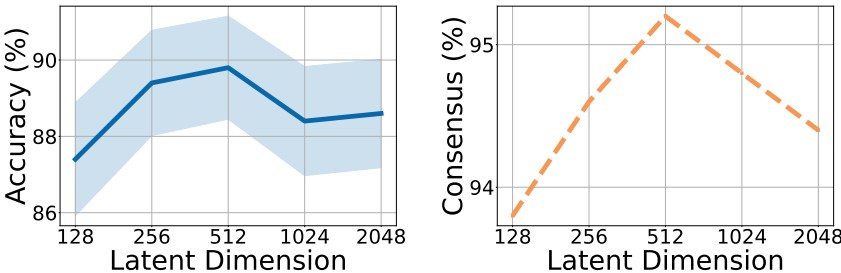

Figure 9: Effect of varying latent dimensionality on MATH for Qwen-3-1.7B [Yang et al., 2025]. A similar trend is observed, confirming that the benefits of higher latent capacity generalize across architectures.

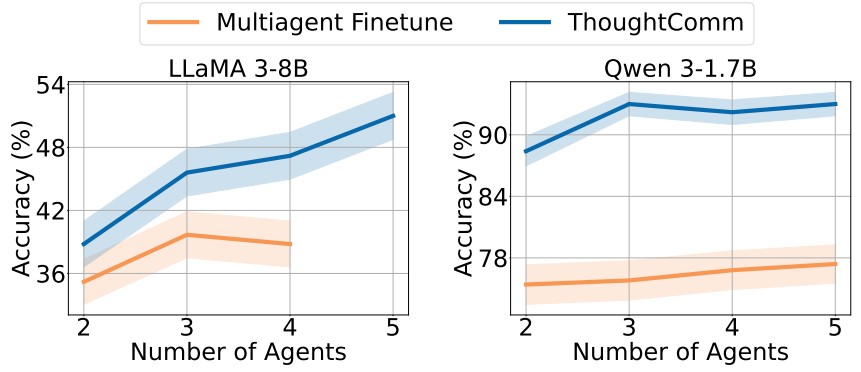

Figure 10: Performance as the number of agents increases on MATH for Llama-3-8B-Instruct and Qwen-3-1.7B. The missing data point is due to runtime limit exceeded.

## C.4 Additional Results on Varying Number of Agents

We next analyze how increasing the number of collaborating agents influences performance. All experiments are conducted with two rounds, latent dimension of 1024, and a single prefix token on the MATH dataset.

As shown in Fig. 10, both models initially benefit from more agents, achieving notable gains when increasing from 2 to 3. However, beyond 3 agents, accuracy plateaus or slightly declines, particularly for the Multiagent Finetune baseline. In contrast, THOUGHTCOMM maintains stable accuracy even as the number of agents grows, highlighting its robustness to redundant or conflicting signals.

