# OpenReview forum: "Thought Communication in Multiagent Collaboration"
_NeurIPS.cc/2025/Conference — NeurIPS 2025 spotlight_

### Official Review · Reviewer_R1aq · 2025-06-03

**Clarity:** 2
**Significance:** 3
**Originality:** 4
**Rating:** 5
**Confidence:** 4

**Summary:**

This paper proposes enhancing the performance of LLM-based multi-agent collaborative discussions by introducing shared and private hidden thoughts. Specifically, the authors design an autoencoder with an assignment matrix to encode the latent thoughts from the last token embeddings. These latent thoughts are then split into shared and private components based on the assignment matrix and delivered to each agent in the next discussion round via prefix-tuning. The effectiveness of the assignment matrix design is first validated through experiments on a synthetic dataset, and the proposed method is further evaluated on two mathematical reasoning benchmarks, demonstrating its effectiveness.

**Questions:**

1. Why is it necessary to group latent thoughts by agreement level?

2. Is $w_{\alpha_j}$ manually set or learned during training?

3. How can we validate that the learned structure of thoughts is meaningful in math reasoning tasks?

4. Have you considered a baseline without any structure of thoughts? Specifically, what happens if all last-token embeddings from previous agents are directly used as prefixes for the next round, without grouping or assigning structure?

5. This looks very similar to Latent Dirichelet Allocation. Have you tried to use a latent generative model directly on natural language in a similar way to LDA? The latent generative model is not bound to hidden representation. In my opinion, the design principle of THOUGHTCOMM is based on latent thoughts, not latent embeddings. Latent thoughts can also be tokens.

6. Can you provide a concrete case study of THOUGHTCOMM in action? Is there an example showing intermediate natural language outputs for each round of discussion?

7. Do latent thoughts offer interpretable dimensions in real-world tasks like math reasoning? The toy example in Figure 1 includes interpretable concepts like "carrying luggage" and "punctuality". Can similar concepts be extracted in real-world tasks?

8. Does scaling up the structure of thoughts lead to continued performance gains?

9. From the example in Figure 1, the THOUGHTCOMM design aligns well with scenarios where the problem is approached from different perspectives. However, in cases where multiple agents hold opposing views and need to debate and persuade each other, can THOUGHTCOMM still operate effectively?

**Ethical Concerns:**

["NO or VERY MINOR ethics concerns only"]

**Final Justification:**

The authors have addressed my concerns regarding additional experiments and interpretability. I am willing to raise my rating to 5.

**Limitations:**

yes

**Paper Formatting Concerns:**

No issue.

**Quality:**

3

**Strengths And Weaknesses:**

Strengths

1. The paper explores an interesting and meaningful direction: multi-agent communication beyond natural language.
2. It adopts an efficient training strategy that avoids heavy computation overhead.
3. The design is intuitively appealing—communication as a composition of private and shared perspectives aligns well with how collaborative reasoning may work.
4. Experiments are conducted on diverse and up-to-date model backends, showing the method's applicability across different LLMs.

Weaknesses

1. The experimental scale is limited, involving only three agents. It remains unclear how the method generalizes to more complex multi-agent settings.
2. The experimental analysis lacks depth. For instance, the paper focuses on tuning parameters like prefix length and debate rounds, which are arguably not central to the proposed THOUGHTCOMM framework. More critical variables such as the scale of latent thoughts and the number of agents (which directly relate to the structure of thoughts) deserve closer examination.
3. The implementation details are overly brief. The main paper and appendix provide only three sentences on implementation. Key details regarding the autoencoder, prefix-finetuning hyperparameters, and training setup are missing.

---

> ### Author Rebuttal · Authors · 2025-07-31
>
> Thank you very much for your thoughtful and detailed feedback. In light of it, we have incorporated **additional experiments** and **new discussions** in the updated version. Please find our point-by-point responses below.
>
> In all tables, M denotes Multiagent Finetuning, T denotes ThoughtComm.
>
> **Q1:** It remains unclear how the method generalizes to more complex multi-agent settings.
>
> **A1:** We sincerely appreciate your constructive suggestions. Accordingly, we have conducted **new experiments varying the number of agents** on MATH:
>
> |Base Model|# Agents|Method|Accuracy ($\uparrow$)|
> |----------|--------|------|---------------------|
> |LLaMA 3-8B|2|M|0.3520$\pm$0.0214|
> |||T|0.3880$\pm$0.0218|
> ||3|M|0.3968$\pm$0.0219|
> |||T|0.4560$\pm$0.0223|
> ||4|M|0.3880$\pm$0.0218|
> |||T|0.4720$\pm$0.0223|
> ||5|M|Running|
> |||T|0.5100$\pm$0.0224|
> |Qwen 3-1.7B|2|M|0.7540$\pm$0.0193|
> |||T|0.8840$\pm$0.0143|
> ||3|M|0.7580$\pm$0.0192|
> |||T|0.9300$\pm$0.0114|
> ||4|M|0.7680$\pm$0.0189|
> |||T|0.9220$\pm$0.0120|
> ||5|M|0.7740$\pm$0.0187|
> |||T|0.9300$\pm$0.0114|
>
> As expected, performance **improves with more agents** and outperforms SOTA.
>
> ---
>
> **Q2:** More critical variables such as the scale of latent thoughts and the number of agents deserve closer examination.
>
> **A2:** In light of your great suggestions, we have conducted **additional experiments** on both the **scale of latent thoughts (below)** and the **number of agents (table in A1)**. For the latent thought scales, we vary the latent dimension and evaluate on MATH with two agents.
>
> |Base Model|Dim|Accuracy ($\uparrow$)|
> |----------|----|----------------------|
> |LLaMA 3-8B|128|0.3840$\pm$0.0218|
> ||256|0.4160$\pm$0.0220|
> ||512|0.4160$\pm$0.0220|
> ||1024|0.3880$\pm$0.0218|
> ||2048|0.4100$\pm$0.0220|
> |Qwen-3-1.7B|128|0.8740$\pm$0.0148|
> ||256|0.8940$\pm$0.0138|
> ||512|0.8980$\pm$0.0135|
> ||1024|0.8840$\pm$0.0143|
> ||2048|0.8860$\pm$0.0142|
>
> We observe that increasing the latent dimension improves performance up to a point. This is expected: too few latents under-represent latent thoughts, while too many introduce noise and the curse of dimensionality.
>
> ---
>
> **Q3:** More details of the implementation.
>
> **A3:** Thanks for your constructive feedback. Accordingly, we have added details as follows:
>
> > “We use a standard autoencoder with $\ell_1$ sparsity on latent-observed connections. The latent dimension is $1024$. For prefix-finetuning, we use one prefix token and generate one sentence. Training uses batch size 128 and learning rate 1e-5. We extract states from 2/3 model depth.”
>
> ---
>
> **Q4:** Why is it necessary to group latent thoughts by agreement level?
>
> **A4:** Thank you for the thoughtful question. Agreement level helps uncover latent thought hierarchy: shared thoughts reflect common knowledge or shared premises, while private ones capture individual reasoning. This separation helps recovering the thought structure, helping agents to adatively consider different types of latent thoughts.
>
> ---
>
> **Q5:** Is $w_{\alpha_j}$ manually set or learned?
>
> **A5:** Thanks for the reminder. It is learned during training. We have highlighted this in the updated manuscript where it first appears (Line 256).
>
> ---
>
> **Q6:** Meaningfulness of learned structure in math reasoning.
>
> **A6:** In light of your insightful question, we have conducted **interpretation analysis** on the recovered latent structured thoughts in MATH. We find that **concepts and structures similar to Figure 1 can be extracted in real-world math reasoning tasks**.
>
> We follow the automatic interpretability methods commonly used in mechanistic interpretability literature. Specifically, for a target latent variable, we identify the top-10 samples (questions) with the highest activation, and prompt GPT-4o to summarize the semantic pattern across those cases given the conversation contexts. Below are a few representative examples:
>
> > **Shared concepts:**
> >
> > - "Utilizing trigonometric identities to simplify trigonometric expressions."
> >
> > - “Using arithmetic operations to manipulate fractions and decimals”
> >
> > - **Insight:** These are all basic mathematical knowledge that are shared in most problems, and thus are relevant to all agents.
> >
> > **Private concepts:**
> >
> >
> > - **Agent 1’s concept:** “finding the least common multiple (LCM) by identifying the highest power of each prime number that appears in the factors of given integers”
> >
> > - **Insight:** Other agents may opt for different methods to find the LCM, such as listing multiples or using a simpler approach.
> >
> > - **Agent 2’s concept:** “using the quadratic formula over factoring when solving quadratic equations”
> >
> > - **Insight:** This technique differentiates Agent 1 from others who might opt to factor the quadratic expression instead.
> >
> > - **Agent 3’s concept:** “using Heron's Formula to calculate the area of triangles when given the lengths of all three sides.”
> >
> > - **Insight:** While other agents might use basic geometric formulas or the base-height method for calculating the area of triangles, Agent 2 specifically opts for Heron's Formula, indicating a distinctive computational preference for this method in relevant scenarios.
>
> In summary, the latent thoughts **indeed encode concepts needed for tasks**, and identifying the dependency structure is important given the **fundamental difference between shared and private thoughts**
>
> ---
>
> **Q7:** Consider a baseline without any structure of thoughts.
>
> **A7:** Thanks, we have conducted **new experiments** accordingly. All other settings are the same as the main experiments in Table 1. The performance drop supports the effect of considering structure.
>
> |Base Model|Method|Accuracy ($\uparrow$)|
> |----------|------|---------------------|
> |LLaMA 3-8B|Single Answer|0.3620$\pm$0.0215|
> ||M|0.3968$\pm$0.0219|
> ||T|0.4560$\pm$0.0223|
> ||T w/o Structure|0.4380$\pm$0.0222|
> |Qwen 3-1.7B|Single Answer|0.4360$\pm$0.0222|
> ||M|0.7580$\pm$0.0192|
> ||T|0.9300$\pm$0.0114|
> ||T w/o Structure|0.9180$\pm$0.0123|
>
> ---
>
> **Q8.** Discussion on using a latent generative model directly on natural language in a similar way to LDA.
>
> **A8.** This is a great idea. We haven’t tried LDA specifically, but we discussed a similar idea of **extracting latent thoughts directly from text** in **L386-389** and **Appendix C**. This can sidestep the need to access model states, thus generalizing the framework to **fully closed-source** LLMs. Inspired by your suggestion, we have also highlighted LDA as a promising alternative choice.
>
> ---
>
> **Q9:** Can you provide a concrete case study of ThoughtComm including language conversation?
>
> **A9:** Thanks for your great suggestion. Here is a concrete example. Due to the length limit, we omitted some duplicated content with clear notes.
>
> **Summary:**  Agents began with different answers and **converged on the correct answer**. Specifically, the answer of **Agent 2** has been corrected by itself after communication. The other agents **noted** the wrong answer in Agent 2, but were **not misled** by it.
>
> **Question:** In how many ways can 13 students be selected from a group of 15 students to serve on a chalkboard-cleaning team?
>
> - **Ground-truth answer:** 105.
>
> **First round:**
>
> - **Agent 1:**
>
>   >To solve this problem, we can use the combination formula:
>   >
>   >C(n, k) = n! / (k!(n-k)!)
>   >
>   >where n is the total number of students (15) and k is the number of students selected (13). Plugging in the values, we get:
>   >
>   >C(15, 13) = 15! / (13!(15-13)!)
>   >
>   >= 15! / (13!2!)
>   >
>   >= (15 * 14 * 13!)/((13 * 12)*13!)
>   >
>   >= 15 * 14/2
>   >
>   >= 105
>   >
>   >(Omitted)
>   >
>   >Answer: \boxed{105}
>
> - **Agent 2:**
>
>   >(Omitted)
>   >
>   >C(15, 13) = 15! / (13!(15-13)!)
>   >
>   >= 15! / (13!2!)
>   >
>   >= (15 × 14 × 13 × 12 × 11 × 10 × 9 × 8 × 7 × 6 × 5 × 4 × 3 × 2 × 1) / ((13 × 12 × 11 × 10 × 9 × 8 × 7 × 6 × 5 × 4 × 3 × 2 × 1)(2 × 1))
>   >
>   >= 3003
>   >
>   >(Omitted)
>   >
>   >So, the answer is:
>   >
>   >\boxed{3003}
>
> - **Agent 3:** (Omitted, the same solution as Agent 1’s)
>
>
> **Second round:**
>
> - **Agent 1:**
>
>   >Here is the updated bullet point summary:
>   >
>   > (Omitted, repeated the previous solution)
>   >
>   >• Some agents simplify the expression further by canceling out the common factors, resulting in C(15, 13) = 3003, while others simplify it to C(15, 13) = 15 × 14 / 2 = 105.
>   >
>   >• Regardless of the simplification, the correct answer is that there are 105 ways to select 13 students from a group of 15 to serve on a chalkboard-cleaning team.
>   >
>   >Answer: \boxed{105}
>
> - **Agent 2:**
>
>   >(Omitted)
>   >
>   >• The formula is simplified by cancelling out the common factors in the numerator and denominator, and we get:
>   >
>   >C(15, 13)=15! / (13!(15-13)!)
>   >
>   >= 15! / (13!2!)
>   >
>   >= (15 × 14 × 13!)/((13 × 12)*13!)
>   >
>   >= 15 × 14 / 2
>   >
>   >(Omitted)
>   >
>   >Answer: \boxed{105}
>
> - **Agent 3:** (Omitted, the same solution as Agent 1’s)
>
> ---
>
> **Q10:** Does scaling up the structure of thoughts lead to continued performance gains?
>
> **A10:** Yes, we observe continued performance improvements as we scale up. Specifically, we conducted additional experiments varying both the number of agents and the dimensionality of latent thoughts (see **tables in A1 and A2**), confirming the benefits of scaling in our framework.
>
> ---
>
> **Q11:** In cases where multiple agents hold opposing views and need to debate and persuade each other, can the method still operate effectively?
>
> **A11:** Thank you for raising this important point. Our framework is **explicitly designed to support disagreement and resolution among agents**. It identifies and preserves private thoughts to respect opposing views while leveraging shared latent structure to facilitate convergence. This is supported both theoretically and empirically. As illustrated in our case study (see **A9**), agents with **initially conflicting answers** are able to **exchange latent thoughts** and **ultimately reach the correct consensus**.

---

> > ### Comment · Reviewer_R1aq · 2025-08-03
> >
> > Thank you for your response. It effectively addresses my concerns regarding the additional experiments and interpretability. I am willing to raise my rating to 5.

---

> > > ### Author Response · Authors · 2025-08-04
> > >
> > > We sincerely appreciate your support and insightful suggestions! Thanks again for your time and effort reviewing our paper.

---

### Official Review · Reviewer_d5cT · 2025-07-02

**Clarity:** 2
**Significance:** 4
**Originality:** 3
**Rating:** 5
**Confidence:** 3

**Summary:**

This paper proposes a novel communication paradigm for multi-agent LLM systems called thought communication, which replaces traditional natural language exchanges with direct interaction via shared latent variables. The authors formalize the notion of thoughts as latent variables and present a nonparametric latent variable model that allows agents to share and reason over these latent constructs. The core theoretical contribution is a set of identifiability results proving that shared and private thoughts can be recovered under mild conditions. Building on this, the authors develop THOUGHTCOMM, a framework leveraging a sparsity-regularized autoencoder and prefix adaptation for direct latent communication among agents. Extensive experiments on synthetic data and real-world reasoning benchmarks MATH, GSM8K demonstrate substantial performance gains over standard message-passing approaches.

**Questions:**

Please refer to the weakness.

**Ethical Concerns:**

["NO or VERY MINOR ethics concerns only"]

**Final Justification:**

The authors have successfully addressed my concerns with new experiments and analysis. In this respect, I think this paper can contribute to the community, and I have raised the acceptance score.

**Limitations:**

yes

**Quality:**

3

**Strengths And Weaknesses:**

Strength:

1.	The paper presents an interesting and well-explained idea: letting agents share their internal thoughts directly instead of talking through natural language, which can be vague or unclear. This tackles a well-known problem in how AI agents work together and offers a new way for them to communicate more effectively.

2.	The paper establishes rigorous identifiability results for both shared and private latent thoughts ensures that the recovered latent representations reflect the true internal structure of agent reasoning.

3.	The experiments results show clear improvements in both accuracy and agent consensus over existing baselines and are consistent across model scales.

4.	The method avoids full LLM finetuning and works with frozen models, which makes it applicable in real settings.

Weakness:
1.	The proposed framework assumes access to agent model states, which may not be available for closed-source models. While acknowledged as a limitation, the practical impact is non-trivial, especially given the paper’s ambitious real-world aspirations.

2.	Although the theory ensures identifiability, there’s limited discussion on whether the recovered latent thoughts are interpretable or actionable by humans or other systems. Some empirical visualization or case study would have strengthened the narrative.

3.	The paper adopts a sparsity-regularized autoencoder as its core estimation engine. However, there’s little exploration of its architecture, training sensitivity, or how design choices impact identifiability or downstream performance.

4.	The evaluation is focused on math reasoning, which, while difficult, represents a narrow task distribution. It’s unclear how well THOUGHTCOMM performs in domains like planning, negotiation, or open-domain dialogue.

---

> ### Author Rebuttal · Authors · 2025-07-31
>
> We truly appreciate your insightful and constructive feedback. Based on it, we have performed **new experiments** and **additional discussions** to the updated manuscript. Below, we provide detailed responses to each point.
>
> ---
>
> **Q1:** The proposed framework assumes access to agent model states, which may not be available for closed-source models. While acknowledged as a limitation, the practical impact is non-trivial, especially given the paper’s ambitious real-world aspirations.
>
>
> **A1:** Thanks for raising this insightful point. As noted in **Lines 386–389** and detailed in **Appendix C**, even for **fully closed-source models**, latent thought communication remains applicable. This is because the latent representation need not rely strictly on access to internal model states. Instead, we can directly encode latent thoughts from the summary of the generated text itself, **bypassing the need for model states entirely**. While constructing effective summarization representation from text is itself a separate nontrivial task, this alternative approach offers a practical and extensible solution when access to open-source models is restricted. The details are referenced as follows:
>
>
> > "**Alternative to model states.** Our main framework assumes access to the model states $H_t$ of each agent before communication. These states provide a rich representation of the agent’s processing of context and are used as inputs to our autoencoder for recovering latent thoughts. However, such internal states may be inaccessible in many practical settings, particularly when using closed-source or API-restricted models.
>
> > In these cases, a viable alternative is to replace the model state $H_t^{(i)}$ of each agent with a compact embedding extracted from its textual response. Specifically, one can apply a context-aware embedding model to summarize the agent’s generated text into a fixed-size vector, which is then treated as a proxy for the unavailable model state.
>
> > Crucially, this embedding does not need to preserve any structure among agents, nor does it need to reflect the agent’s intent. Its only requirement is to provide a compressed summary of the textual content at the linguistic level. Examples of such embedding methods include those from models like BERT or RoBERTa, pooled sentence embeddings from Sentence-BERT, or output vectors from instruction-tuned embedding APIs. These methods are designed to produce compact, semantically meaningful vectors that summarize the surface content of a given text.
>
> > Once such an embedding is obtained for each agent, the rest of the framework remains unchanged. The collection of response embeddings is treated as a surrogate for $H_t$ and passed through the sparsity-regularized autoencoder to recover latent thoughts $\hat{Z}_t$. From that point on, latent communication proceeds identically: inferring shared/private thoughts, routing them based on recovered structure, and injecting them into agents via prefix adaptation.
>
> > This replacement provides a drop-in mechanism to support latent communication in scenarios where model internals are inaccessible, enabling broader applicability of the framework across both open- and closed-source agents."
>
> Thanks again for your great question. We have further highlighted it in Section 4. Please feel free to let us know if there are any further questions.
>
> ---
>
>
> **Q2:** Although the theory ensures identifiability, there’s limited discussion on whether the recovered latent thoughts are interpretable or actionable by humans or other systems. Some empirical visualization or case study would have strengthened the narrative.
>
>
> **A2:** We sincerely appreciate your constructive suggestions. We fully agree with you that some case studies could be very helpful. In light of it, we carried out a **new interpretability analysis of the latent thoughts**, inspired by standard techniques in mechanistic interpretability. Specifically, for a target latent variable, we identify the top-10 samples (questions) with the highest activation, and prompt GPT-4o to summarize the semantic pattern across those cases given the corresponding conversation contexts. Below are a few representative examples illustrating shared thoughts across agents, as well as agent-specific private reasoning:
>
> > **Shared concepts:**
> >
> > - "Utilizing trigonometric identities to simplify trigonometric expressions."
> >
> > - “Using arithmetic operations to manipulate fractions and decimals”
> >
> > - **Insight:** These are all basic mathematical knowledge that are shared in most problems, and thus are relevant to all agents.
> >
> > **Private concepts:**
> >
> >
> > - **Agent 1’s concept:** “finding the least common multiple (LCM) by identifying the highest power of each prime number that appears in the factors of given integers”
> >
> > - **Insight:** Other agents may opt for different methods to find the LCM, such as listing multiples or using a simpler approach.
> >
> > - **Agent 2’s concept:** “using the quadratic formula over factoring when solving quadratic equations”
> >
> > - **Insight:** This technique differentiates Agent 1 from others who might opt to factor the quadratic expression instead.
> >
> > - **Agent 3’s concept:** “using Heron's Formula to calculate the area of triangles when given the lengths of all three sides.”
> >
> > - **Insight:** While other agents might use basic geometric formulas or the base-height method for calculating the area of triangles, Agent 2 specifically opts for Heron's Formula, indicating a distinctive computational preference for this method in relevant scenarios.
>
> To summarize, we can observe that the **latent thoughts indeed encode concepts needed for specific tasks**. Moreover, identifying the dependency structure is important given the **fundamental difference between shared and private thoughts**.
>
> ---
>
> **Q3:** More exploration on the empirical study of the sparsity-regularized autoencoder.
>
>
> **A3:** Thanks so much for the insightful suggestion. Accordingly, we have conducted a set of **new experiments** to study the regularized autoencoder in both the identifiability and down stream performances.
>
> **Identifiability:**
>
> We first evaluate the effect of the $\ell_1$ regularization strength, which promotes sparsity – a key difference from standard autoencoders. Using a hidden dimension of 1024 and 1000 samples, we report the mean correlation coefficient (MCC) across varying sparsity weights:
>
> | Sparsity weight | MCC ($\uparrow$)             |
> |-----------------|------------------------------|
> | $1\mathrm{e}{-5}$  | 0.84$\pm$0.01              |
> | $1\mathrm{e}{-4}$  | 0.85$\pm$0.02              |
> | $1\mathrm{e}{-3}$  | 0.87$\pm$0.01              |
>
> From the results, the performance drops when the sparsity regularization is not strong enough, **supporting the necessity of sparsity regularization**.
>
> **Downstream Performance:**
>
> We then vary the latent dimensionality of the autoencoder and evaluate multi-agent accuracy on the MATH dataset using two different LLM backbones. The number of agents is set to two, and other settings follow those used in Table 1 of the main paper.
>
> | Base Model     | Dim  | Accuracy ($\uparrow$)       | Consensus ($\uparrow$) |
> |----------------|------|------------------------------|-------------------------|
> | LLaMA 3-8B      | 128  | 0.3840$\pm$0.0218          | 0.6870                  |
> |                | 256  | 0.4160$\pm$0.0220          | 0.6880                  |
> |                | 512  | 0.4160$\pm$0.0220          | 0.6800                  |
> |                | 1024 | 0.3880$\pm$0.0218          | 0.6867                  |
> |                | 2048 | 0.4100$\pm$0.0220$         | 0.7020                  |
> | Qwen-3-1.7B     | 128  | 0.8740$\pm$0.0148          | 0.9380                  |
> |                | 256  | 0.8940$\pm$0.0138          | 0.9460                  |
> |                | 512  | 0.8980$\pm$0.0135          | 0.9520                  |
> |                | 1024 | 0.8840$\pm$0.0143          | 0.9480                  |
> |                | 2048 | 0.8860$\pm$0.0142          | 0.9440                  |
>
> We observe that increasing the latent dimension improves performance up to a point, after which the gains plateau or slightly degrade. This is **expected**: while too few latent variables can under-represent the thought space, excessively large dimensions may introduce unnecessary noise in addition to the curse of dimensionality.
>
> Thank you again for your suggestion to extend the empirical analysis. We hope these new results provide greater clarity into the behavior of the regularized autoencoder. Please feel free to let us know if you have further questions or feedback.
>
> ---
>
> **Q4:** The evaluation is focused on math reasoning, which, while difficult, represents a narrow task distribution. It’s unclear how well ThoughtComm performs in domains like planning, negotiation, or open-domain dialogue.
>
> **A4:** Thank you for the thoughtful suggestion. Our current focus on math reasoning tasks is deliberate: these datasets require **a combination of skills**, including **multi-step logical reasoning, collaborative language use, implicit planning**, and **negotiation between agents**. In this sense, they serve as a challenging testbed that integrates multiple reasoning and interaction skills, rather than isolating a single capability. This may also be why they have become standard benchmarks in prior work.
>
> That said, we agree that evaluating performance on domains like planning, negotiation, and open-domain dialogue would further strengthen the case for generality. We have highlighted this in the revised discussion and see this as an important direction for future work. Please feel free to let us know if you have any further suggestions; we sincerely appreciate it.

---

> > ### Author Response · Authors · 2025-08-06
> > **Appreciation and Check-In Before Discussion Ends**
> >
> > Thank you again for your time and valuable feedback. In response to your insightful suggestions, we have added the suggested experiments and expanded discussions in the updated version. As the discussion phase is nearing its end, please do not hesitate to let us know if there are any further questions.

---

> ### Comment · Reviewer_d5cT · 2025-08-07
>
> Thank you for your response. I really appreciate your addressing my concerns and revising the manuscript. I'll increase my rating to 5.

---

> > ### Author Response · Authors · 2025-08-07
> > **We sincerely appreciate your support**
> >
> > Thank you sincerely for your support. We greatly appreciate the time and care you devoted to reviewing our paper. Your insightful feedback and encouragement have been genuinely meaningful to us.

---

### Official Review · Reviewer_J712 · 2025-07-03

**Clarity:** 2
**Significance:** 2
**Originality:** 3
**Rating:** 4
**Confidence:** 3

**Summary:**

This paper proposes to leverage extracted thoughts (i.e. representation vectors) of one model that can be injected into another model in order to improve results.

**Questions:**

I am struggling with why the results on Table 1 are actually that good. How can the collaboration of just three 0.6B models substantially outperform an 8B models? What is the intuition behind this? Is the knowledge actually in the trained adapter, so is most of the improvement not about the multi-agent but about the finetuning?

Figure 4: " The red line marks the red line threshold typically considered identifiable when exceeded." => what does "typically considered identifiable" mean?

**Ethical Concerns:**

["NO or VERY MINOR ethics concerns only"]

**Final Justification:**

In light of the rebuttal, I opted to improve my rating for this paper. Specifically adding the proof sketches will help the readibility of the paper.

**Limitations:**

Limitations should be discussed in more depth. Also a proper discussion of and positioning with respect to related work is missing.

**Paper Formatting Concerns:**

none.

**Quality:**

3

**Strengths And Weaknesses:**

Overall, this paper pursues a very interesting idea in an up-and-comming hot topic, i.e., multi-agent systems. It is mostly well written and nicely motivated. However, the presentation has still some issues, which leaves a couple of questions open to me.

While reading, I struggled with the connection between the data generating process in the problem formulation and how to connect it to LLMs. The authors should consider making this more explicit.

The reported results are actually impressively good. So good that I actually have trouble to understand how this is achieved. The collaboration of 3 agents of the same type should not be able to solve a problem that much better than a single one.

Related work to the novel approach is not properly discussed. For example, how does the extraction of thoughts relate to research on interpretability, specifically to probing or sparse autoencoders?
Furthermore, the proof in the appendix should be explained in more detail; to me if was incomprehensible.
Unfortunately, the full code is not availbale for review.

---

> ### Author Rebuttal · Authors · 2025-07-31
>
> Thank you very much for your thoughtful and detailed feedback. In response, we have included **additional experiments** and added **new discussions** to the revised version. Please see our detailed point-by-point replies below.
>
> ---
>
> **Q1:** More discussion on the connection between the data generating process and LLMs.
>
> **A1:** Thank you for this helpful suggestion. In light of it, we have added the following paragraphs in the updated manuscript:
>
> >  “When generating responses, LLMs operate on internal representations learned during training. We model this by assuming that each agent’s model state (e.g., hidden state $H$ before token generation) is produced from a set of latent thoughts $Z$ via an unknown function. Specifically, we posit that $H$ is generated from $Z$ through a complex, nonparametric function $f$, capturing the model’s internal reasoning process.
>
> > This view aligns with the data-generating perspective adopted in recent mechanistic interpretability work (e.g., Olah et al., 2020; Cunningham et al., 2023), where model states are treated as outputs of latent concepts. Our setting differs in two important ways: (1) we assume a nonparametric, potentially highly nonlinear mapping $f$, whereas sparse autoencoders typically use a sparse linear decoder; and (2) we focus on multi-agent systems, where identifying structured dependencies between agents and their latent thoughts is essential for collaborative reasoning beyond language.”
>
> We hope this could make the connection clearer. We appreciate your insight and would be glad to elaborate further if helpful.
>
> ---
>
> **Q2:** What’s the intuition behind the strong results? It’s surprising that 3 agents of the same type can so significantly outperform a single one. How can collaboration among three 0.6B models outperform an 8B model? Does most of the gain come from the trained adapter rather than the multi-agent setup?
>
>
>
> **A2:** Thank you for raising these important and insightful questions. It has been consistently observed in recent literature that multi-agent collaboration and debate can lead to substantial performance gains compared to a single agent [1]. Similarly, increasing the number of agents in our framework also leads to improved results, consistent with this line of research.
>
> That said, our goal is not to re-establish the benefits of multi-agent communication, which are now well-verified in the literature. Rather, our work seeks to **advance the mode of multi-agent communication itself** – from surface-level language to **structured latent thoughts**. The core contribution is to show that communication through latent thoughts can further improve multi-agent reasoning. This is supported by both theory (identifiability of latent thoughts and structure) and experiments: as shown in Table 1 of the manuscript, our method **consistently outperforms the existing SOTA** (Multiagent Finetuning [1]) under the **same number of agents and rounds**, demonstrating the added value of latent thought communication beyond pure language.
>
> Meanwhile, one may also wonder whether the improvements over the previous multi-agent SOTA come at the cost of increased computation. To address this, we conducted an **additional runtime comparison** using LLaMA 3-8B-Instruct on the MATH dataset, measuring wall-clock runtime as the number of agents varies:
>
> | # Agents | Multiagent Finetuning (s) | ThoughtComm (s) |
> |----------|----------------------------|------------------|
> | 4        | 15520.44                   | 6027.23          |
> | 5        | 22263.76                   | 7867.38          |
> | 6        | 24755.92                   | 9633.05          |
>
>
> These results show that our method not only achieves **higher accuracy** but does so with **significantly lower time**. This indicates that the performance gains are not due to increased compute. In fact, they suggest that latent communication may offer a **more efficient route to improved coordination**.
>
> In summary:
>
> - **Compared to single-agent baselines**, most of the improvement comes from multi-agent collaboration (as expected).
>
>
> - **Compared to prior multi-agent methods**, most of the gain comes from latent communication, not from increased computation or additional test-time scaling.
>
> Regarding the strong performance of Qwen 0.6B, we note that even in the single-agent setting, it slightly outperforms some 8B models on MATH. This phenomenon has also been observed in prior work and likely reflects differences in the domain specialization of models.
>
> Thanks again for your constructive feedback. We appreciate your careful reading and hope the introduced discussions and experiments could be helpful.
>
> ---
>
> **Q3:** More discussion on the connection with interpretability research, proof intuitions, and implementation.
>
> **A3:** We sincerely appreciate your suggestions. We fully agree with you that the connection with mechanistic interpretability (e.g., sparse autoencoder) should be further highlighted. In light of this, we have **added the following paragraphs** in the revised manuscript:
>
> > “Recent work in mechanistic interpretability has explored structured latent representations, notably through sparse autoencoders (SAEs), which aim to discover interpretable features from LLM hidden states [Olah et al., 2020; Cunningham et al., 2023]. These approaches treat model states as being generated from latent concepts and recover sparse linear encodings for interpretability.
>
> > Our work aligns with this perspective in spirit: both aim to uncover the latent generative process behind model hidden states. However, there are two key differences. (1) SAEs assume a sparse **linear** relationship between latent variables and observed states, whereas our formulation allows for a **general nonlinear** generative process $f$, providing greater flexibility and applicability to complex real-world settings. (2) The goal of SAEs is primarily interpretability – interpreting the underlying generative features – whereas our focus is on **enabling latent thought communication**. We recover latent thoughts and their dependency structures not only for interpretability, but to directly support efficient multi-agent reasoning. Empirically, this latent communication has been shown to outperform language-based exchange in collaborative tasks.”
>
> Moreover, regarding the proofs, we have now included **proof sketches for all theorems** in the main paper to provide a direct intuition underlying the theoretical framework.
>
> > “**Proof sketch of Theorem 1:** Since we make no assumptions on the functional form of $f$, the latent variable distribution, or access to auxiliary information (e.g., interventions or multiple domains), the only available signal for identification lies in the structural relationship between latent and observed variables. In the nonparametric setting, this structure is captured by the non-zero pattern of the Jacobian of the generating function $f$. We require that the Jacobian spans its support subspace, ensuring that the population adequately reflects the underlying dependency structure. Leveraging this, we show that the intersection of latent variables influencing any pair of observed variables can be disentangled from all other components.”
>
> > “**Proof sketch of Theorem 2:** Similar to Theorem 1, we begin by excluding degenerate cases where the population fails to span the Jacobian's support subspace. We then analyze the Jacobian's structure using linear algebra, focusing on the structural distinction between complements and intersections of latent supports, which enables the disentanglement of private latent variables unique to each observed variable in a pair.”
>
> >”**Proof sketch of Theorem 3:** We begin by connecting the support structure of the Jacobian to the latent-observed dependency graph, under the span assumption. We then show that the support of the estimated Jacobian must contain that of the true Jacobian. By incorporating a sparsity regularization, we eliminate superfluous entries and reduce the indeterminacy, allowing us to identify the dependency structure up to a permutation of the latent dimensions.”
>
> Due to policy restrictions, we cannot include external links in the rebuttal, but we would like to highlight the implementation details as follows. As mentioned in Line 757, we build on the original codebase of Multiagent Finetuning [1]. The main additional module introduced in our framework is the sparsity-regularized autoencoder, which is a standard autoencoder trained with an $\ell_1$ penalty on the Jacobian to encourage sparse structural dependencies. The adapter for prefix injection is implemented following established techniques from prefix-tuning [3], and both components are lightweight and modular, requiring only the hidden states of the LLMs as input. We have expanded Appendix D with additional architectural and training details.
>
> Thanks again for your thoughtful feedback. Please let us know if you have any additional questions.
>
> ---
>
> **Q4:** Clarification of "typically considered identifiable".
>
> **A4:** Thanks for your great question. In the context of nonparametric identifiability, a high mean correlation coefficient (MCC) between estimated and ground-truth latent variables is widely used as a proxy for identifiability [2]. When the MCC is high, each estimated latent variable tends to correspond to a distinct ground-truth one, suggesting **one-to-one recovery** and thus identifiability. In contrast, **non-identifiable settings typically yield MCC values closer to 0.3**, reflecting entangled or overlapping representations. We have clarified this point in the revised manuscript to avoid potential ambiguity.
>
> ---
>
> [1] Subramaniam et al. "Multiagent Finetuning: Self Improvement with Diverse Reasoning Chains."
>
> [2] Hyvärinen et al. “Identifiability of latent-variable and structural-equation models: from linear to nonlinear.”
>
> [3] Li and Liang. “Prefix-Tuning: Optimizing Continuous Prompts for Generation.”

---

> > ### Author Response · Authors · 2025-08-06
> > **Thanks for Your Review: Follow-up Before Discussion Ends**
> >
> > Thank you again for your time and thoughtful feedback. In light of your insightful suggestions, we have added the suggested experiments and expanded the discussions accordingly. As the discussion period is coming to a close, please let us know if there are any additional questions.

---

> > > ### Author Response · Authors · 2025-08-07
> > > **Follow-up Before Discussion Concludes**
> > >
> > > Dear Reviewer J712,
> > >
> > > Thanks again for your time and effort reviewing our manuscript. We understand you may be very busy during this period, but just wanted to briefly check in before the discussion phase ends, in case there are any additional questions or clarifications we could assist with. We would be more than glad to discuss.
> > >
> > > We sincerely appreciate your time and consideration.
> > >
> > > Thanks,
> > >
> > > Authors

---

> > ### Comment · Reviewer_J712 · 2025-08-08
> >
> > Dear Authors,
> >
> > I thank you for your thorough reply and considering my comments. I think the proposed adatations to the script will certainly improve the paper. Considering your rebuttal (to my as well as to my co-reviewers comments), I will adjust my rating by one step.

---

> > > ### Author Response · Authors · 2025-08-08
> > >
> > > Dear Reviewer J712,
> > >
> > > Thank you so much for going through the rebuttals and taking them into considerations when re-evaluating our paper. We are deeply grateful to your constructive feedback and score adjustment!
> > >
> > > Best regards,
> > >
> > > Authors

---

### Official Review · Reviewer_rt7c · 2025-07-04

**Clarity:** 2
**Significance:** 3
**Originality:** 3
**Rating:** 5
**Confidence:** 4

**Summary:**

The paper introduces thought communication, a paradigm where large-language-model agents exchange latent representations nstead of natural-language messages to reduce ambiguity in multi-agent coordination. It formalizes agent states as outputs of an unknown injective function applied to latent thoughts and proves that shared thoughts, private thoughts, and their agent-specific structure are identifiable under mild assumptions. Building on this theory, the authors propose THOUGHTCOMM: a sparsity-regularized autoencoder extracts latent thoughts from agents’ hidden states, and lightweight prefix adapters inject the relevant thoughts back into each agent during generation. Experiments on synthetic data validate the identifiability guarantees, while tests on MATH and GSM8K show accuracy and consensus gains over single-agent baselines and multi-agent finetuning, with minimal additional training overhead.

**Questions:**

1) This form of multiagent approach is a form of test-time scaling since it involves both several rounds of debate and it involves querying several language models. Because of this, it seems unfair to compare against "single answer" that involves no test-time scaling. Do you think the experiments improvements could just results from the increase in test-time scaling? I would really like to see an attempt at giving each of the approaches similar amounts of test-time compute for a fair comparison. For examples, approaches like using thinking models or top-k sampling.

2) It's claimed multiple times that the approach is model-agnostic. As it stands the claim is unsupported – it would be great to see some experimental results backing this up, or rephrasing that this is speculation.

3) It's quite surprising that varying the prefix length doesn't affect the models' performance at all. I would think too small of a prefix length would bottleneck how much the models could communicate but this doesn't seem to be the case. What do you think is going on here?

**Ethical Concerns:**

["NO or VERY MINOR ethics concerns only"]

**Final Justification:**

I believe the inclusion of these extra experiments and clarification in the final version will strengthen the paper. The two strongest reasons (in order) that convinced me to increase my score are: (1) the new mechanistic interpretability experiments and (2) the wall-clock test time experiments.

**Limitations:**

The limitations of the approach could be more thoroughly discussed. For example, multi-agent appraoces introduce additional overhead in terms of test-time scaling. Also, having models communicated through thought vectors rather than natural language introduces risks in making the models much less interpretable and black-box. Communication through natural language at least allows human oversight into the models. I would really like to see a discussion of this point in the paper.

**Paper Formatting Concerns:**

minor: 153: identifiabilty → identifiability

**Quality:**

3

**Strengths And Weaknesses:**

Strengths:
- The paper demonstrates strong empirical improvements over existing methods in well-established benchmarks.
- Introduces a novel “thought communication” paradigm and backs it with formal identifiability proofs. These results are theoretically interesting and provide a strong basis for the method’s empirical performance.
- The paper is clearly written and well organized.

Weaknesses:
- The experimental comparison isn’t fair in terms of test-time compute and I think the baselines could be stronger. See questions.
- There is a bit of a disconnect between the theory an empirical experiments. My understanding is that the real-world experiments use a dependency structure such that all messages are shared to all agents. It would be nice to have some real-world demonstration of why selective sharing of messages is important.
- I think the paper could really benefit from a more detailed analysis of what the thought vectors are learning to exchange between agents. It’s not clear to me whether the empirical improvement just comes from an increase in test-time compute, rather than the models actually engaging in a meaningful form of collaboration. I think an analysis of the contents of the thought vectors could provide a lot of insight in this direction. Some experiments in this direction would likely lead me to improve my score.
- Some of the claims are a bit bold and I think distract from the main points of the paper: “We hope this work marks a potential step toward superhuman collaboration, which is essential for superhuman intelligence”

---

> ### Author Rebuttal · Authors · 2025-07-31
>
> We sincerely appreciate your constructive feedback and insightful comments. In light of these, we have conducted **new experiments** and introduced **new discussions** in the updated version. Please see our point-by-point responses below:
>
> ---
>
> **Q1:**  It seems unfair to compare against the “single answer” baseline, which involves no test-time scaling. Could the improvement just result from the increase in scaling? An attempt to evaluate with the consideration of compute would be great.
>
> **A1:** Thank you for the insightful comment. While we agree that the single-agent baseline does not control for test-time scaling, **our primary comparison is with Multiagent Finetuning** [1], a **SOTA in multi-agent communication** that also uses **multiple agents and communication rounds**. In Table 1 of the submission, ThoughtComm consistently outperforms Multiagent Finetuning across settings, highlighting the added benefit of latent thought communication beyond language. The single-answer baseline serves mainly as a sanity check, since prior work in this line of research has already shown that multi-agent debate improves reasoning significantly.
>
> Motivated by your suggestion, we further conducted an **additional runtime comparison** using LLaMA 3-8B-Instruct on MATH, measuring wall-clock runtime on two H100 GPUs as the number of agents varies:
>
> | # Agents | Multiagent Finetuning (s) | ThoughtComm (s) |
> |----------|----------------------------|------------------|
> | 4        | 15520.44                   | 6027.23          |
> | 5        | 22263.76                   | 7867.38          |
> | 6        | 24755.92                   | 9633.05          |
>
> These results show that **our method achieves higher performance with significantly lower cost**, indicating that the gains do not stem from increased compute. In fact, this suggests that latent communication may be a **more efficient route to improved coordination**. Thanks again for your constructive insight. We hope the new results could shed light on the potential of going beyond language, both in performance and efficiency.
>
> ---
>
> **Q2:** My understanding is that the real-world experiments use a dependency structure such that all messages are shared to all agents. It would be nice to have some real-world demonstration of why selective sharing of messages is important.
>
>
> **A2:** Thank you for raising this point. It is correct that, in our current experiments, all text messages are visible to all agents. However, our focus is **not on message-level filtering**, but on **selective sharing of latent thoughts** – the internal representations that generate those messages. This selective sharing occurs in the latent space, where for each agent, we infer a personalized subset of latent thoughts based on recovered structural dependencies and the identified latent thought vector. These determine which thoughts influence an agent’s internal state, enabling agents to consider only relevant cognitive content while ignoring unrelated noises. In this sense, although messages are broadcast, the latent thoughts driving coordination are filtered and routed in a personalized manner, capturing differences in roles, priors, or goals. We believe this is a more faithful model of how effective collaboration should work.
>
> In light of this, we have added the following discussion in the updated manuscript to further highlight the distinction:
>
> > “In our real-world experiments, the recovered dependency structure plays a critical role in shaping how latent thoughts are routed to each agent. After extracting the shared latent space via the sparsity-regularized autoencoder, we apply a structural mask to ensure that each agent only receives the latent dimensions that are relevant to its own internal representation. This filtering directly affects how the injected prefixes are constructed for each agent during the next round of generation. Although the surface-level messages are broadcast, the actual content used to condition each agent’s reasoning is selectively constructed in the latent space, reflecting the personalized structure of shared and private thoughts.”
>
> ---
>
> **Q3**: It’s not clear to me whether the empirical improvement just comes from an increase in test-time compute, rather than the models actually engaging in a meaningful form of collaboration. I think an analysis of the contents of the thought vectors could provide a lot of insight in this direction. Some experiments in this direction would likely lead me to improve my score.
>
>
> **A3:** We sincerely appreciate your insightful suggestions. Accordingly, we have conducted **additional experiments** on both the **computation** and **interpretation of latent thoughts**.
>
> The results on the computation cost have been shown in detail in **A1**, where our method **achieves higher performance with significantly lower cost**, **compared to SOTA** in multi-agent collaboration and debate.
>
> For interpretability, we performed a new analysis inspired by the mechanistic interpretability literature. For each latent variable, we identified the top-10 most activated samples (questions) on MATH and used GPT-4o to summarize the shared semantic pattern based on the corresponding conversation contexts. A few representative examples are shown below.
>
>
> > **Shared concepts:**
> >
> > - "Utilizing trigonometric identities to simplify trigonometric expressions."
> >
> > - “Using arithmetic operations to manipulate fractions and decimals”
> >
> > - **Insight:** These are all basic mathematical knowledge that are shared in most problems, and thus are relevant to all agents.
> >
> > **Private concepts:**
> >
> >
> > - **Agent 1’s concept:** “finding the least common multiple (LCM) by identifying the highest power of each prime number that appears in the factors of given integers”
> >
> > - **Insight:** Other agents may opt for different methods to find the LCM, such as listing multiples or using a simpler approach.
> >
> > - **Agent 2’s concept:** “using the quadratic formula over factoring when solving quadratic equations”
> >
> > - **Insight:** This technique differentiates Agent 1 from others who might opt to factor the quadratic expression instead.
> >
> > - **Agent 3’s concept:** “using Heron's Formula to calculate the area of triangles when given the lengths of all three sides.”
> >
> > - **Insight:** While other agents might use basic geometric formulas or the base-height method for calculating the area of triangles, Agent 2 specifically opts for Heron's Formula, indicating a distinctive computational preference for this method in relevant scenarios.
>
>
> In conclusion, **the latent thoughts indeed encode concepts needed for specific tasks**. Moreover, identifying the dependency structure is important given the **fundamental difference between shared and private thought**. Thanks again for your very constructive feedback.
>
> ---
>
> **Q4:** Some of the claims are a bit bold and I think they distract from the main points of the paper.
>
> **A4:** Indeed, these sentences may have been overly aspirational in tone. Thank you for pointing this out. We have removed these sentences (e.g., L19 and L390) in the updated manuscript.
>
> ---
>
> **Q5:** Why is it claimed that the approach is model-agnostic?
>
> **A5:** Thank you for highlighting this point. By “model-agnostic,” we mean that for any given set of LLMs, a sparsity-regularized autoencoder can be applied to reconstruct their latent thoughts based on the concatenated model states. Once trained, the same autoencoder can be used across communication rounds without further model-specific tuning.
>
> That said, we acknowledge that this requires access to open-source models to get the model states. So while our framework does not rely on architectural details of the LLMs themselves, it is not completely plug-and-play in the strictest sense.
>
> Interestingly, **even without knowing any information on the language model, it is still possible to apply our framework**. This is because we can always directly encode latent representation from the text of responses, and thus bypass the model's hidden state, which generalizes the applicability to completely **closed-source** models. At the same time, finding a good summarization representation of text is a separate topic. This was mentioned in **L386-389** and had been detailed in **Appendix C**.
> Thanks again for your great question, which provides us with the opportunity to further clarify it. We have removed the term ‘model-agnostic’ and highlighted the above discussion to avoid any potential confusion.
>
> ---
>
> **Q6:** It's quite surprising that varying the prefix length doesn't affect the models' performance at all. I would think too small a prefix length would bottleneck how much the models could communicate, but this doesn't seem to be the case. What do you think is going on here?
>
>
> **A6:** Thank you for raising this thoughtful observation – we had a similar reaction initially. In some cases, we do observe slight performance gains as the prefix length increases, but the effect is not consistent across models or tasks. A possible explanation lies in the distinction between a token embedding and a prefix embedding. While both have the same dimensionality (e.g., 1024), a token embedding is tied to a single vocabulary item and typically encodes the semantics of just that **one discrete token**, often lying on a lower-dimensional subspace. In contrast, a prefix embedding is a free parameter optimized to encode **many continuous latent thoughts**, leveraging the full capacity of the embedding space. As a result, even a single prefix token can convey a rich bundle of cognitive content, making longer prefixes less essential in our setting. This may explain the diminishing returns with increased prefix length.
>
> ---
>
> [1] Subramaniam et al. "Multiagent Finetuning: Self Improvement with Diverse Reasoning Chains." ICLR 2025

---

> > ### Comment · Reviewer_rt7c · 2025-08-06
> >
> > Thank you for the detailed response. I believe the inclusion of these extra experiments and clarification in the final version will strengthen the paper. I am happy to increase my score accordingly.

---

> > > ### Author Response · Authors · 2025-08-06
> > > **Thanks so much for your support**
> > >
> > > Thanks so much for your support! We sincerely appreciate your time and effort in reviewing our paper. Your suggestions and encouragement mean a lot to us.

---

### Note · Authors · 2025-08-12

Dear Reviewers, AC, and SAC,

We sincerely thank you for the time, effort, and thoughtful feedback you have devoted to the review and discussion, which has brought many constructive suggestions that will undoubtedly help further strengthen the manuscript. We would like to take this opportunity to briefly summarize the fruitful exchange of ideas:

---

**Contributions**

- **Conceptually**, we introduce thought communication, a new paradigm for multi-agent LLMs that models communication through the latent thoughts underlying agents’ reasoning, formalized via a general latent generative model.

- **Theoretically**, we establish nonparametric identifiability guarantees for both shared and private latent thoughts and the full thought–agent structure, without side information or restrictive functional constraints.

- **Practically**, we develop *ThoughtComm*, a principled framework that extracts and selectively shares latent thoughts among agents, yielding consistent gains across diverse LLMs, tasks, and settings.

---

**Rebuttal**

During the rebuttal phase, in light of the reviewers’ insightful suggestions, we conducted additional experiments and expanded several discussions. We are deeply grateful for these constructive inputs and are glad that all questions have been properly addressed, as acknowledged by all reviewers.

- **Additional experiments**: runtime comparison; varying sparsity weight, latent dimensionality, and number of agents; ablation on the structure of thoughts.


- **Further analysis**: interpretability study on the semantics of learned thoughts and a case study of the communication procedure.


- **Expanded discussions**: importance of selective sharing; connection between the data-generating process and LLMs; links to mechanistic interpretability; proof sketches for all theorems; and several additional clarifications.
---

We truly appreciate your support, guidance, and engagement throughout the process. Your feedback has been instrumental in advancing the potential of thought communication, and we are grateful for the opportunity to learn from it.

Warm regards,

Authors of Submission 3474

---

### Decision · Program_Chairs · 2025-09-17

**Decision:**

Accept (spotlight)

**Comment:**

The paper introduces a new communication paradigm for multi-agent LLM named thought communication. Instead of communicating via natural language, the authors propose to enable agents to communicate directly through latent representations (thoughts). The authors prove that shared and private latent thoughts (and their agent-specific structure) are identifiable under mild assumptions. The authors propose ThoughtComm, using a sparsity-regularized autoencoder to extract latent thoughts and prefix adaptation to inject thoughts back to agents.

The reviewers recognize the novelty of the thought communication, its theoretical analysis and strong emprical results. While certain concerns were raised such as limited evaluation settings and missing analysis, the authors provided additional experimental result and clarifications in rebuttal. After rebuttal, the reviewers are unanimously positive about the paper. Thus, the recommendation is acceptance.